# Artificial hibernation/life-protective state induced by thiazoline-related innate fear odors

Tomohiko Matsuo[1,3], Tomoko Isosaka[1,3], Lijun Tang[1], Tomoyoshi Soga[2], Reiko Kobayakawa[1✉] &
Ko Kobayakawa [1✉]

Innate fear intimately connects to the life preservation in crises, although this relationships is not fully understood. Here, we report that presentation of a supernormal innate fear inducer 2-methyl-2-thiazoline (2MT), but not learned fear stimuli, induced robust systemic hypothermia/hypometabolism and suppressed aerobic metabolism via phosphorylation of pyruvate dehydrogenase, thereby enabling long-term survival in a lethal hypoxic environment. These responses exerted potent therapeutic effects in cutaneous and cerebral ischemia/reperfusion injury models. In contrast to hibernation, 2MT stimulation accelerated glucose uptake in the brain and suppressed oxygen saturation in the blood. Whole-brain mapping and chemogenetic activation revealed that the sensory representation of 2MT orchestrates physiological responses via brain stem Sp5/NST to midbrain PBN pathway. 2MT, as a supernormal stimulus of innate fear, induced exaggerated, latent life-protective effects in mice. If this system is preserved in humans, it may be utilized to give rise to a new field: "sensory medicine."

[1] Institute of Biomedical Science, Kansai Medical University, Osaka 573-1010, Japan. [2] Institute for Advanced Biosciences, Keio University, Tsuruoka, Yamagata 997-0052, Japan. [3] These authors contributed equally: Tomohiko Matsuo, Tomoko Isosaka. ✉email: kobayakr@hirakata.kmu.ac.jp; kobayakk@hirakata.kmu.ac.jp

Hibernating animals have the ability to survive under the state of hypothermia/hypometabolism[1], and also have resistance to ischemia/reperfusion (I/R) injury[2]. Therefore, if artificial hibernation can be induced for non-hibernators like humans, it is expected that irreversible brain damage due to the arrest of blood flow can be reduced[3]. It is well known that therapeutic hypothermia exerts protective effects on brain function[4]. However, physical cooling methods alone cannot resolve the contradiction that arises between the reduction of body temperature mediated by external cooling systems and heat generation induced by homeostasis to maintain body temperature[5].

Mice brain has an intrinsic hypothermia-inducing system. Single night fasting or calorie restriction for a few days at a standard ambient temperature can induce torpor in mice[6]. Pacap-positive neurons in the medial preoptic area (MPA) are activated in the torpor state, and artificial activation of these cells induces hypothermia[7–9]. It is pointed out that such a hypothermia-inducing system in the brain could be applied to emergency medicine. For this purpose, sensory stimuli that can rapidly drive the intrinsic hypothermia-inducing system are useful.

We hypothesized that organisms have evolved an undiscovered latent life-protective mode characterized by hypothermia/hypometabolism, which can be induced by the brain in life-threatening situations. In order to demonstrate this idea, we need a technology to induce crisis perception in model animals and humans. Fear is evoked when the brain perceives life-threatening danger, having evolved to induce behavioral and physiological responses that increase the survival chances of individuals[10–12]. However, protective effects conferred by fear remain unclear. Elucidating these effects is important not only for understanding the evolution of fear but also for using these potential protective effects in medical applications.

Fear stimuli induce a wide variety of physiological responses. The presentation of conditioned fear stimuli leads to increases in heart rate and body temperature in mice[13]. In contrast, the heart rate is decreased by 50% when patients with phobias are exposed to their phobic stimuli[14]. Fear is induced by innate and learned mechanisms[11,15]. We previously demonstrated that innate and learned fear information is integrated antagonistically in the fear center of the brain to regulate a hierarchical relationship in which innate fear behaviors are prioritized over learned fear behaviors[16]. Assuming that behavioral responses induced by innate and learned fears are antagonistically regulated, the physiological responses induced by these two types of fear emotions may also be antagonistic. However, the absence of effective stimuli to induce innate fear in animal models is a major obstacle in the attempts to clarify physiological responses induced by innate fear. Furthermore, the mechanism by which physiological responses induced by innate fear stimuli contribute to the generation of bioprotective effects is mostly elusive.

Early ethological studies revealed that innate behaviors are induced more robustly by artificial, exaggerated stimuli (i.e. supernormal stimuli) than by natural stimuli. For example, chicks will peck a stick with an exaggerated version of the shape and color of their parents' beak, more frequently than the real ones[17]. Predator odorants, for example, 2,4,5-trimethyl-3-thiazoline (TMT), a fox secretion, and cat collars, induce innate fear responses in rodents[18,19]. However, fearful behaviors induced by these odorants are much weaker than learned fear responses induced by conditioned odorants previously paired with electric foot shocks (FSs)[16]. Therefore, we have optimized the chemical structure of TMT to develop artificial thiazoline-related fear odors (tFOs) with more than ten times greater activities to induce innate freezing behavior compared to any other previously identified innate fear odors[16,20,21]. We also demonstrated that tFOs bind to transient receptor potential ankyrin type1 (TRPA1)

receptor protein in trigeminal nerves to induce fearful behaviors[21]. TFOs such as 2-methyl-2-thiazoline (2MT) work as unique supernormal stimuli to induce the most robust innate fear responses (e.g., freezing behavior) in mice compared to any other known sensory stimuli. Thus, we hypothesized that utilizing tFOs would allow us to discover latent life-protective effects intrinsic to an innate fear that could not be uncovered using previous experimental models. Indeed, we used tFOs to decipher biological significances of innate fear and successfully discovered a series of unexpected protective effects intrinsic to an innate fear that determine survivability in various aspects.

## Results

**Antagonistic physiological responses between innate and learned fear.** Fear can be quantitatively measured based on indices, represented by freezing behavior, serum levels of stress hormones, and neck muscle electromyography[22–25]. Innate fear induced by 2MT and learned fear induced by anisole (Anis) previously paired with FSs were comparable based on known fear indices[16]. However, we observed a clear difference in cutaneous temperature induced by these two types of fear. Mice subjected to innate fear stimuli, but not those subjected to learned fear stimuli, exhibited an ~3 °C drop in cutaneous body temperature along the spine (Figs. 1A, B and Supplementary Movie 1). Fear is described as "spine-chilling" in various languages and innate fear matches this expression. Next, we analyzed core body temperature and heart rate using an implantable telemetry system. Core body temperature was also decreased by ~3 °C in response to innate fear stimuli. In contrast, core body temperature was increased by ~0.5 °C following exposure to learned fear stimuli (Fig. 1C). Interestingly and importantly, long exposure to 2MT led to a decrease in body temperature to near-ambient temperature after ~5 h and an almost complete drop in locomotor activity after ~2 h (Fig. 1E and Supplementary Fig. 1), resembling the state of torpor, a transient hibernation-like state[26]. Thereafter, upon 2MT removal, body temperature recovered, and mice behaved normally (Fig. 1F). While learned fear stimuli induced only slight increases in heart rate, innate fear stimuli caused robust changes (up to 50% reduction) within a few minutes, in accordance with the physiological responses observed in phobia patients[14] (Fig. 1D).

In addition to electric shock, intraperitoneal (IP) injection of lithium chloride (LiCl) can also be used as an unconditioned stimulus[27]. Thus, IP injection of LiCl may also elicit innate fear/stress responses. Indeed, LiCl administration itself elicited decreases in both core body temperature and heart rate (Figs. 1G, H). Conversely, presentation of learned fear stimuli previously paired with LiCl injection increased core body temperature (Fig. 1G). Subsequently, we examined whether other innate fear stimuli also induce hypothermia and bradycardia. Restraint in a tight space is considered to induce innate fear because it prevents access to food and water and may lead to death. Our results indicated that innate fear induced by restraint also resulted in decreased core body temperature and heart rate (Figs. 1I, J). These results demonstrate that, regardless of stimulus type, innate fear stimuli lead to decreases in body temperature, while learned fear stimuli induce increases in body temperature. Thus, we propose a model that fear is not a single emotional state; however, there are at least two distinct fear-related states: innate hypothermic fear state and learned hyperthermic fear state. We then aimed to elucidate the mechanisms responsible for the characteristic hypothermia induced by innate fear stimuli.

**Innate fear confers hypoxic resistance.** Decreases in body temperature can be achieved either via the promotion of heat exchange at the body surface or via inhibition of heat production.

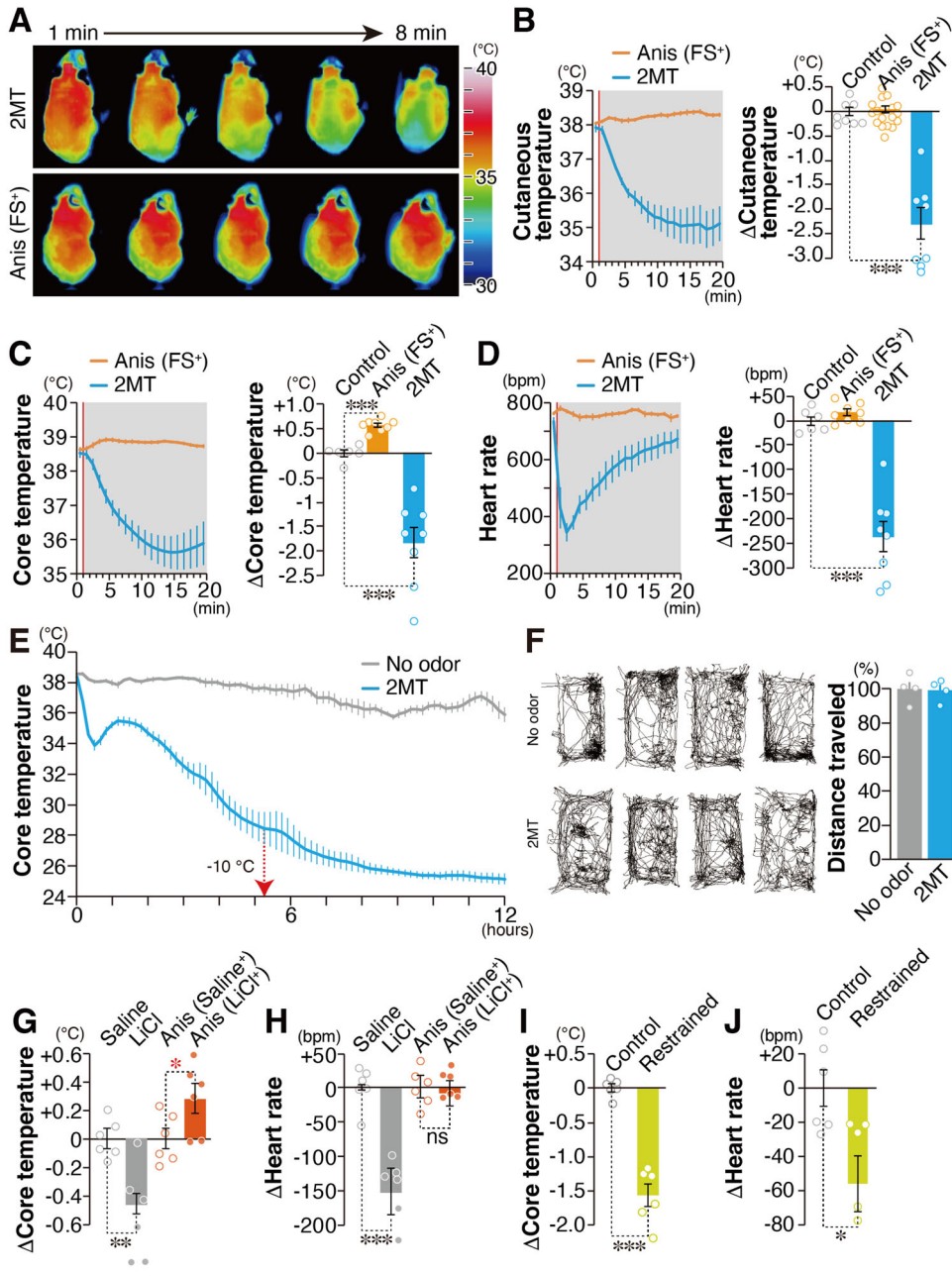

**Fig. 1 Physiological responses induced by innate versus learned fear stimuli. A** Infrared dorsal view images taken 1–8 min after odorant presentation. **B**–**D** Temporal analyses (left panels) and the mean values (right panels) for cutaneous temperatures (**B**; $n = 8$ for control, $n = 18$ for Anis (FS+), and $n = 8$ for 2MT), core body temperatures [**C**; $n = 6$ for control, $n = 8$ for Anis (FS+), and $n = 8$ for 2MT], and heart rates (**D**; $n = 6$ for control, $n = 8$ for Anis (FS+), and $n = 8$ for 2MT) in response to a conditioned odor (anisole) paired with foot shock [Anis (FS+); orange] and 2MT (blue). Red vertical lines indicate the onset of odor presentation. **E** Temporal analyses of core body temperature in response to long exposure to 2MT and no-odor control ($n = 6$ each). Filter paper scented with 2MT or saline was introduced into the test cage with a lid at 10 min. Core body temperature decreased by >10 °C after 5 h of the odor presentation (red arrow) and reached near-ambient temperature (~2 °C above ambient temperature) after 12 h. Decreasing body temperature to 2–3 °C above ambient temperature is observed in torpor[26]. **F** Filter paper scented with 2MT or saline was introduced into the cage, which was then covered by a plastic wrap for 6 h. After 1 week of recovery, mice were transferred into a new cage and locomotor activities were analyzed ($n = 4$ each). Track plots of the movement (left) and the mean distance traveled (right) are shown. **G**, **H** The mean change in core body temperature (**G**) and heart rate (**H**) in response to intraperitoneal (IP) injection of saline or LiCl (gray) and those induced by a conditioned odorant (anisole) paired with IP injection of saline or LiCl [Anis (Saline+) or Anis (LiCl+); orange ($n = 6$ each)]. **I**, **J** The mean change in core body temperature (**I**) and heart rate (**J**) in response to 5 min of restraint (yellow) and control condition ($n = 6$ each). Data are means ± SEM. One-way ANOVA followed by Dunnett's multiple comparison or Student's $t$ test was performed. *$P < 0.05$; ***$p < 0.01$; ***$p < 0.001$; n.s. not significant.

Heat exchange is achieved by increasing peripheral blood flow. Therefore, we measured peripheral blood flow using laser Doppler blood flowmetry and observed that innate fear stimuli suppressed peripheral blood flow more strongly than did learned fear stimuli ($p = 0.0013$, unpaired Student's $t$ test; Fig. 2A). This result suggests that hypothermia induced by 2MT may be caused by the inhibition of heat production. In mice, brown adipose tissue (BAT) has a major contribution to body temperature homeostasis, and uncoupling protein 1 (UCP1) plays a crucial role in heat production in BAT[28–30]. Thus, UCP1 inhibition may underlie the hypothermia induced by innate fear stimuli. Under thermoneutral conditions (30 °C), energy expenditure to maintain the body temperature is minimal and UCP1-dependent heat production is mostly suppressed[31,32]. Even under such conditions, 2MT-induced hypothermia (Fig. 2B). Moreover, 2MT-induced hypothermia was also observed in UCP1-knockout (KO) mice (Fig. 2C). These results suggest that 2MT suppresses basic metabolism, which is considered as constant under normal conditions, rather than inhibiting UCP1-mediated heat production. Consistent with this hypothesis, 2MT suppressed respiratory rate, blood oxygen saturation, and oxygen consumption (Fig. 2D–F). Hibernation/torpor does not reduce $O_2$ saturation; however, 2MT stimulation clearly reduced $O_2$ saturation. Under normal conditions, the decreased $O_2$ saturation should cause a hypoxic ventilatory response (HVR). However, in the 2MT-induced hypothermic state, the respiratory rate did not increase despite the decreased $O_2$ saturation. This suggests that 2MT stimulation suppresses HVR. In ground squirrels, oxygen saturation increases during the hibernation phase, but oxygen saturation decreases during the arousal from hibernation[33]. Thus, although 2MT stimulation induces hypothermia, it may induce a physiological state closer to the arousal from hibernation stage than to the induction stage of hibernation.

Our findings demonstrated that 2MT stimulation induced decreases in heart rate and body temperature (Fig. 1) and suppression of systemic oxygen consumption (Fig. 2). Similar responses are observed in crisis situations in the natural world: feigned death is a defensive response induced in prey animals when they are physically restrained by predators, and this response is commonly observed in a wide variety of animals ranging from insects to mammals[34]. Decreases in respiratory rate and oxygen consumption are observed during feigned death in the opossum[35]. During constriction by snakes, rats show reductions in body temperature and heart rate before dying due to inhibited respiration[36]. If prey animals can respond to constriction by reducing oxygen consumption, the probability of survival after constriction and associated hypoxia may increase. We examined this possibility of using an experimental model of hypoxia. Mice died within 20 min in the control condition under a hypoxic environment containing 4% oxygen. Surprisingly, almost all mice survived >30 min under hypoxic conditions when they had previously been subjected to 2MT (Fig. 2G). Such anti-hypoxic effects were also induced by prior exposure to restraint, another type of innate fear stimulus, although these effects were weaker than those induced by 2MT (Supplementary Fig. 2A). Contrary to the aforementioned observations, learned fear stimuli or corticosterone injection did not exert anti-hypoxic effects (Supplementary Fig. 2B, C), suggesting that anti-hypoxia is linked to innate fear stimuli, but not learned fear stimuli or stress hormone secretion. We then determined an effective concentration of 2MT odor gas for anti-hypoxic activity using gas permeater (Fig. 2H). At least prior stimulation with 0.3 p.p.m. 2MT odor for 30 min significantly increased survival time in 4% $O_2$ condition ($p = 0.0054$, unpaired Student's $t$ test with Bonferroni correction). And submaximal effects were obtained by prior stimulation with 10 p.p.m. 2MT odor. In this condition,

0.23 μg/mL 2MT was detected by quantitative GC-MS analysis in mice serum (Supplementary Fig. 3A, B), which is >2000-fold lower than the reported $LD_{50}$ (lethal dose, 50%) of IP injection of 2MT in mice (600 mg/kg). There are at least two possibilities to explain the anti-hypoxic effects induced by 2MT odor stimulation: one is that 2MT odor directly suppressed mitochondrial oxygen metabolism of somatic cells via the blood stream like hydrogen sulfide ($H_2S$) presentation[37,38], and the other is that sensory representation of 2MT exerted protective effects through central brain functions. Administration of 2 mM (0.2 mg/mL) 2MT solution, which corresponds to an ~1000-fold higher concentration than those detected in serum presented with 10 p.p.m. 2MT, had no effect on oxygen consumption rate in somatic cells (Supplementary Fig. 3C–E). Thus, it is possible that 10 p.p.m. 2MT odor stimulation does not directly suppress mitochondrial oxygen metabolism in somatic cells, but activates specific sensory representation in the brain to induce protective effects against hypoxia.

**2MT induces crisis-response metabolism.** Among all organs, the brain is the most sensitive to hypoxia. Thus, it is possible that 2MT presentation causes shifts in brain metabolism, which increase tolerance to lethal hypoxia. To test this possibility, we compared brain metabolite profiles between 2MT-treated and control mice. Oxygen is utilized in the mitochondrial electron transport chain (ETC). The ETC and tricarboxylic acid (TCA) cycle are tightly coordinated and are essential for the efficient production of ATP. TCA cycle is driven by acetyl-CoA, which is produced via glycolysis. Stimulation with 2MT markedly increased levels of glucose (the starting material for glycolysis) in the brain, as well as levels of glucose-6-P and fructose-6-P, which are produced in the subsequent steps of glycolysis (Fig. 3A). In contrast, 2MT stimulation significantly decreased levels of fructose-1,6-BP and dihydroxyacetone phosphate (DHAP), which are downstream intermediates of glycolysis, as well as levels of the TCA cycle intermediates succinate and malate ($p = 0.0081$ for fructose-1.6-BP and $p = 0.032$ for DHAP, unpaired Student's $t$ test; Fig. 3B). Upregulation of cerebral glucose and fructose-6-P suggests two possibilities (1) that glucose uptake was upregulated or (2) that these metabolites accumulated due to inhibition of glycolysis. To determine which possibility is most likely, we performed metabolic flux analysis, in which we administered $^{13}C$-labeled glucose after 5 min of 2MT stimulation and analyzed $^{13}C$-labeled metabolites in the brain after 20 min of 2MT stimulation (Fig. 3C). Among metabolites shown in Fig. 3M, six kinds of $^{13}C$-labeled metabolites were detected. For the metabolites involved in the glycolysis, although $^{13}C$-glucose-6-P, $^{13}C$-fructose-6-P, $^{13}C$-fructose-1,6-BP, $^{13}C$-DHAP, and $^{13}C$-pyruvate were not detected, $^{13}C$-glucose and $^{13}C$-lactate were significantly upregulated by 2MT stimulation ($p = 0.0057$ for $^{13}C$-glucose and $p = 0.033$ for $^{13}C$-lactate, unpaired Student's $t$ test; Fig. 3D). $^{13}C$-glucose can be present both in the brain and in the blood circulating the brain. Therefore, there are two possibilities: $^{13}C$-glucose was increased in the blood or the incorporation of $^{13}C$-glucose was increased in the brain. However, in addition to $^{13}C$-glucose, $^{13}C$-lactate was significantly increased in the brain compared with that in the control condition, suggesting that the incorporation of $^{13}C$-glucose and glycolysis is enhanced in the brain. The upregulated incorporation of $^{13}C$-glucose, accumulation of glucose-6-P and fructose-6-P, and decreases of fructose-1,6-BP and DHAP can be explained by the suppression of phosphofructokinase; the increased level of $^{13}C$-lactate can be explained by the suppression of pyruvate dehydrogenase (PDH). In this case, fructose-6-P may have been metabolized via other interacting pathways. Alternatively, the increase in $^{13}C$-glucose and $^{13}C$-lactate may have

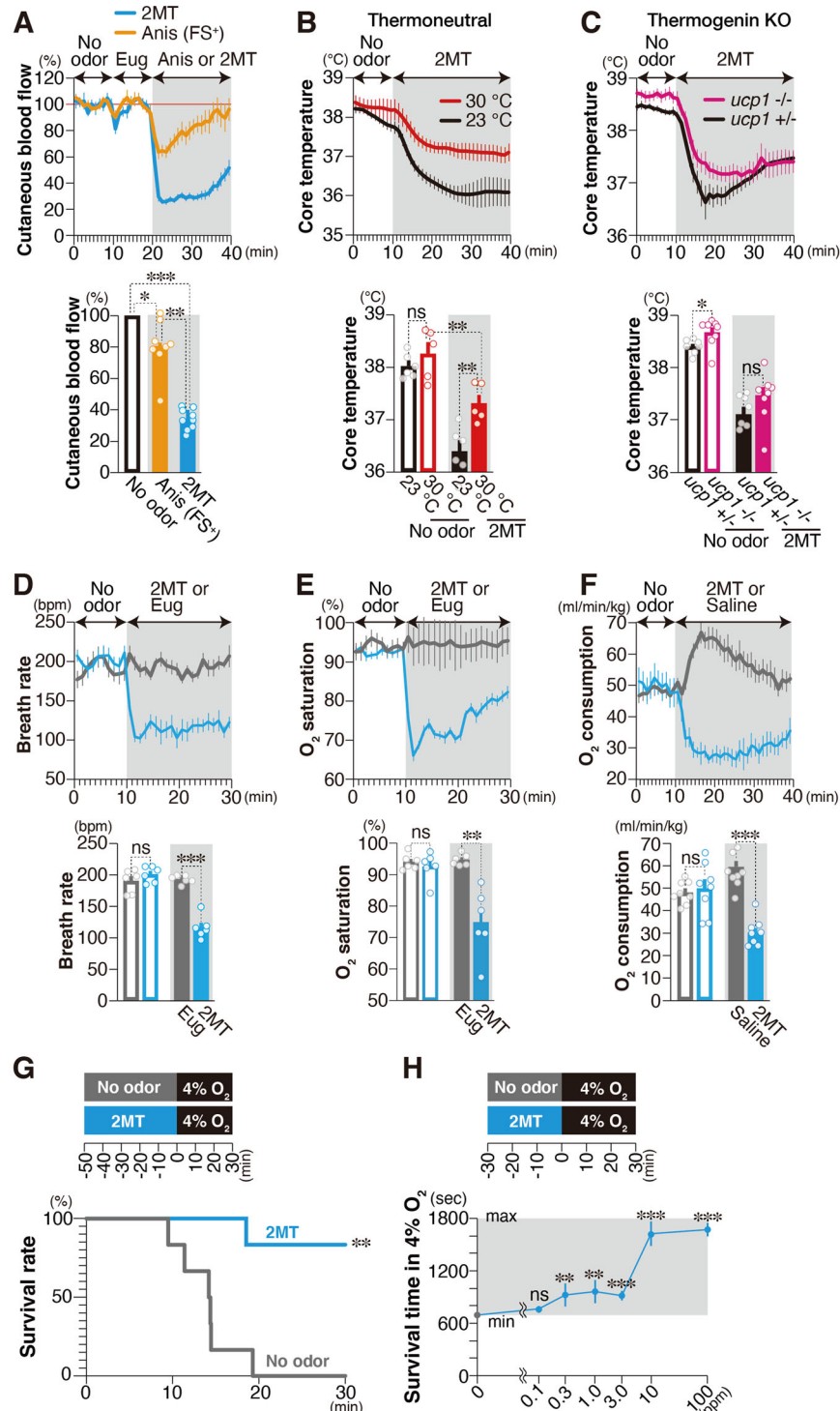

**Fig. 2 Innate fear stimuli suppress basal metabolism to confer anti-hypoxia. A** Temporal analyses of cutaneous blood flow in response to 2MT and Anis (FS+) are shown [$n = 9$ for 2MT and $n = 8$ for Anis (FS+); upper panel]. The mean cutaneous blood flow in the no-odor (black), Anis-FS+ (orange), and 2MT (blue) conditions are also shown (lower panel). Levels in no-odor condition were set at 100%. **B, C** Temporal analyses (upper panels) and mean values (lower panels) for core body temperature induced by 2MT under ambient (black) and thermoneutral (red) conditions (**B**; $n = 6$ for control condition and $n = 5$ for thermoneutral condition), and in control (black) and ucp1$^{-/-}$ (red) mice (**C**; $n = 7$ for control and $n = 8$ for ucp1$^{-/-}$). **D, E** Temporal analyses of respiratory rate (**D**) and oxygen saturation (**E**) in response to eugenol (gray), a neutral odor, and 2MT (blue) ($n = 6$ each). Mean respiratory rate (**D**) and oxygen saturation (**E**) in response to no odor (open bars), eugenol (solid gray bars), and 2MT (solid blue bars) are also shown (lower panels). **F** Temporal analyses of oxygen consumption in response to saline (gray) and 2MT (blue) ($n = 8$ each). Mean oxygen consumption in response to no odor (open bars), saline (solid gray bars), and 2MT (solid red bars) are also shown (lower panel). **G** The experimental procedure (top), and survival rate in 4% oxygen with (blue; $n = 6$) and without (gray; $n = 6$) prior presentation of 2MT. **H** Survival time in 4% oxygen with 30 min prior presentation of indicated concentration of 2MT gas ($n = 5$ for 100 p.p.m., $n = 6$ for other concentrations and $n = 34$ for control). Data are means ± SEM. Student's t test, two-way ANOVA followed by Tukey's multiple comparison or Student's t test followed by Bonferroni correction was performed. For comparison of survival curve, log-rank test was performed. *$P < 0.05$; **$p < 0.01$; ***$p < 0.001$; n.s., $p > 0.05$.

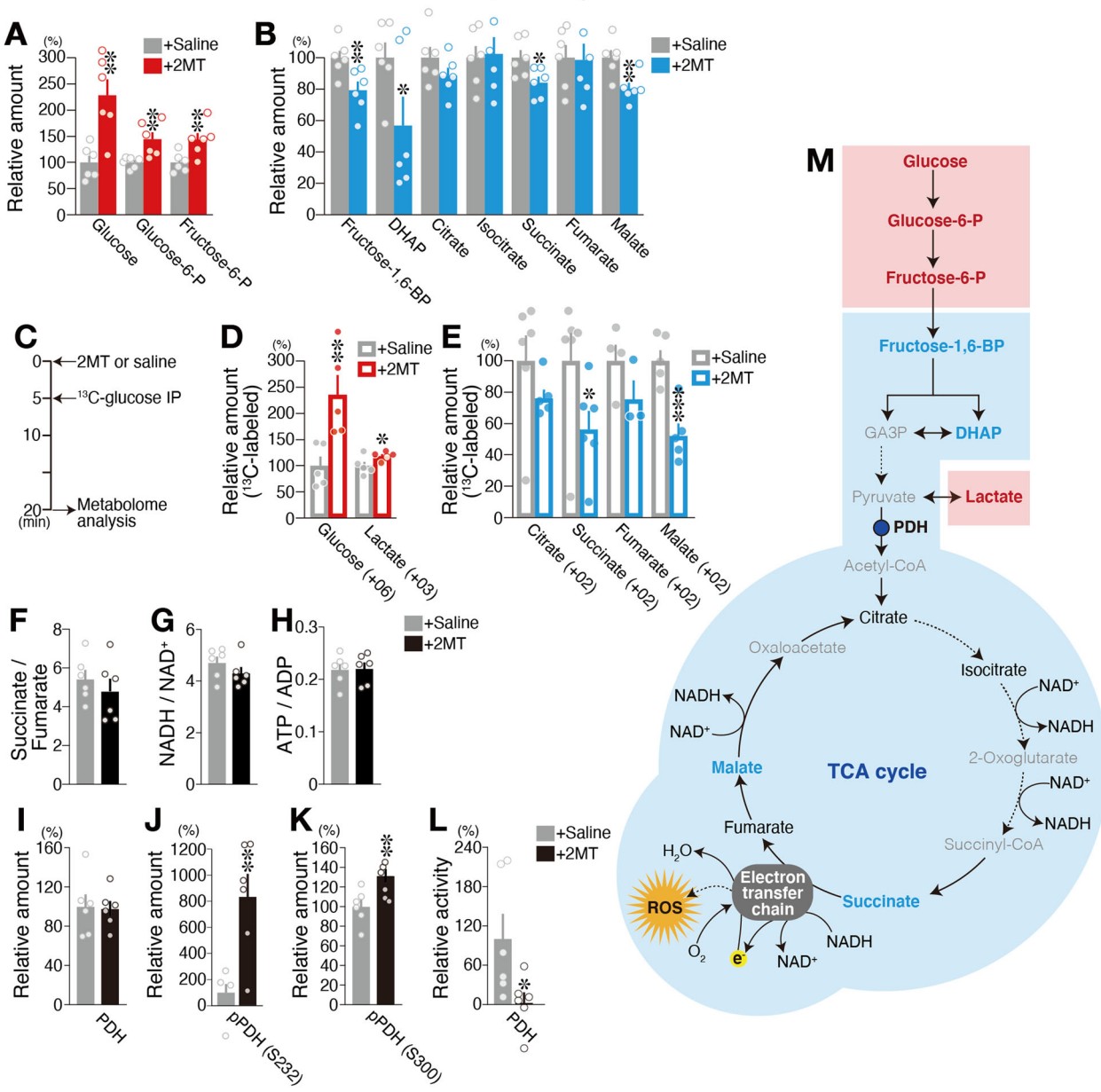

**Fig. 3 Innate fear stimuli induce crisis-response metabolism. A–H** Metabolome analysis for 2MT-treated and control mice ($n = 6$ each). Mean percentages of $^{13}$C-unlabeled (**A**, **B**) and $^{13}$C-labeled (**D**, **E**) metabolites in the brain in response to saline (gray) and 2MT (red or blue). Metabolite levels in response to saline presentation were set at 100%. The experimental procedure (**C**) and the ratios of indicated metabolite pairs are also shown (**F–H**). **I–L** Mean percentages of pyruvate dehydrogenase (PDH) (**I**), pPDH(S232) (**J**), pPDH(S300) (**K**), and PDH activity (**L**) in the brain in response to saline (gray) and 2MT (black). Levels for the saline condition were set at 100% ($n = 6$ each). **M** Schematic diagram of glycolysis and the tricarboxylic acid (TCA) cycle. Red, blue, black, and gray colored are significantly increased, significantly decreased, unchanged, and undetected metabolites, respectively. Data are means±SEM. Student's $t$ test was performed between saline and 2MT conditions. *$P < 0.05$; **$p < 0.01$; ***$p < 0.001$.

been caused by upregulation of glycolysis. Although there is a large amount of blood in the liver, no increase in $^{13}$C-glucose or $^{13}$C-lactate was detected (Supplementary Fig. 4), indicating that the enhancement of glucose uptake is brain-specific. In addition, a previous study reported the results of measuring the metabolites in brain extracts derived from hibernating and active meadow jumping mouse extracts as our study. Glycolytic metabolites such as glucose-6-P and fructose-6-P were markedly decreased in the hibernation phase compared to the active phase, suggesting that glucose uptake is suppressed in the hibernation phase[39]. This result is in contrast to 2MT-induced hypothermia; glycolysis is suppressed to save energy in hibernation, whereas, in 2MT-induced crisis mode, glucose uptake is enhanced probably to protect the brain from crisis. Inhibition of glycolysis has also been reported in the brain of hibernating ground squirrels[40]. In contrast, 2MT stimulation tended to decrease levels of $^{13}$C-citrate, $^{13}$C-succinate, $^{13}$C-fumarate, and $^{13}$C-malate, which are intermediates of the TCA cycle (Fig. 3E). These alterations of metabolites were also not observed in the liver (Supplementary Fig. 4). These results indicate that 2MT stimulation leads to changes in metabolic state in which glucose incorporation is upregulated and aerobic TCA cycle is downregulated in the brain.

Compounds that inhibit the mitochondrial ETC, which mediates ATP synthesis, suppress oxygen consumption. Paradoxically,

because they inhibit the process by which the ETC transfers electrons to oxygen, a large amount of reactive oxygen species (ROS) is generated, which causes irreversible cell damages, leading to cell death[41]. Thus, we examined whether 2MT stimulation inhibits mitochondrial ETC. Inhibition of ETC increases both succinate/fumarate and NADH/NAD$^+$ ratios[42,43]. However, these ratios were not increased (Figs. 3F, G), and the ATP/ADP ratio, which shows cellular energy status, was also not decreased by 2MT stimulation (Fig. 3H). These results indicate that 2MT induces hypoxic metabolism by inhibiting the TCA cycle without aberrant blockade of the ETC. In considering how such a specialized crisis-response metabolism could be induced, we hypothesized it would be effective to suppress the upstream substrate supply for the TCA cycle.

PDH connects glycolysis and the TCA cycle by converting pyruvate to acetyl-CoA (Fig. 3M). In hibernating animals, respiratory rate and oxygen consumption are markedly suppressed, and previous studies have indicated that PDH activity is downregulated during hibernation[44]. If 2MT indeed inhibits PDH activity, it is anticipated to suppress TCA cycle activity, as observed in hibernating animals. Although 2MT stimulation did not alter PDH protein levels in the brain (Fig. 3I), it increased PDH phosphorylation at two phosphorylation sites, both of which are known to suppress PDH activity (Fig. 3J, K). Consistent with this finding, we observed that 2MT stimulation suppressed PDH activity in the brain (Fig. 3L). These results suggest that 2MT stimulation shifts metabolism to a crisis-response mode characterized by the facilitation of glycolysis and suppression of the TCA cycle. This suppression may cause inhibition of ROS production, which could have damaged tissues severely causing individuals' death under hypoxic conditions. There are commonalities between the hibernation state and the 2MT-induced crisis-response mode, such as suppression of the TCA cycle. However, there is also a clear difference: glucose uptake into the brain is greatly suppressed in the hibernation state[45]; on the contrary, it was accelerated in the crisis-response mode (Fig. 3). Thus, it seems that this crisis-response mode is not a passive response caused by hypothermia or oxygen deprivation, but an active response aimed at conferring hypoxia resistance to the brain in a crisis state.

**2MT inhibits I/R injury**. In I/R injury induced by cerebral/myocardial infarction or traumatic injury, excessive ROS generation during reperfusion increases cellular/tissue damage[46]. In crisis-response metabolism induced by innate fear stimuli, mitochondrial ROS generation is likely suppressed via reductions in the activity of the TCA cycle (Fig. 3). It is well known that therapeutic hypothermia exerts protective effects against I/R injury[4]. Thus, 2MT stimulation may exert protective effects against I/R injury by orchestrating the induction of hypothermia and suppressing ROS generation. We examined this possibility using cutaneous and cerebral I/R models.

Cutaneous ischemia was induced by pinching the dorsal skin with ferrite magnetic plates for 12 h, and reperfusion was established by removing them (Fig. 4A). Subsequently, cutaneous ulcer formation was monitored for 9 days. Odor stimulation with 2MT occurred from 30 min prior to 30 min after skin pinching. Ulcer formation was clearly observed in the control condition (Fig. 4B), yet greatly suppressed by 2MT presentation (Figs. 4C, D). Skin pinching in the control condition induced the expression of the apoptosis marker cleaved caspase-3, whereas such expression was attenuated following the presentation of 2MT (Fig. 4E). The expression of β-actin was suppressed in the I/R area in both the control and 2MT conditions. In contrast, 4-hydroxy-2-nonenal (4-HNE), a marker of oxidative stress and lipid peroxidation, was

upregulated in the control condition only (Fig. 4F and Supplementary Fig. 9), indicating that 2MT suppresses the generation of ROS in the I/R injury area.

To determine whether 2MT can alleviate I/R injury when administered not only before but also during reperfusion, thus increasing its therapeutic potential, we developed a mouse model of bilateral common carotid artery occlusion. Both vaporized odor presentation and IP injection of odorant molecules are known to activate sensory neurons[47,48]. Similarly, IP injection of tFO-induced innate fear-related behavioral and physiological responses (Supplementary Fig. 5). Cerebral ischemia was induced by 30 min of bilateral common carotid artery occlusion, following which 2MT was intraperitoneally injected at the time of reperfusion initiation. Two days after reperfusion, we examined cortical infarct size in brain sections (Fig. 4G). Infarct size was smaller in 2MT-treated animals than in saline-treated animals (Fig. 4H–L), suggesting that both 2MT odor presentation and IP administration ameliorate I/R injury.

**Unique features of 2MT-induced hypothermia**. In mice, a hibernation-like state can also be induced by H$_2$S and 2-deoxyglucose[36,45]. H$_2$S acts as an inhibitor of the ETC in the mitochondria and is considered to induce hypothermia/hypometabolism by moderately inhibiting aerobic respiration of somatic cells[36,46]. 2-Deoxyglucose is considered to induce hypometabolism by inhibiting glycolysis[45]. In contrast, tFO does not directly inhibit glucose utilization and function of mitochondrial respiratory chain (Supplementary Fig. 3C–E and Fig. 3). Further, different from 2MT stimulation, the inhalation of H$_2$S does not suppress oxygen saturation in blood[47]. Thus, we speculate that tFOs may induce hypothermia/hypometabolism by a completely different mechanism, in which a sensory representation of tFOs activate an intrinsic crisis-response system in the brain that orchestrates latent life-protective abilities.

Hibernation/torpor is controlled by brain function. However, the present study shows that 2MT stimulation induces several different physiological responses to hibernation/torpor: hibernation/torpor suppresses brain glycolysis[39,40], whereas 2MT stimulation induces the opposite response (Fig. 3). Moreover, oxygen saturation is not reduced by hibernation[33] but is reduced by 2MT stimulation (Fig. 2E). Therefore, it is likely that 2MT stimulation induces a specialized physiological state different from the hibernation state. If this is the case, it suggests that 2MT stimulation induces a different brain activity than that in the hibernation state.

In the torpor entrance stage in the ground squirrels, c-fos messenger RNA (mRNA) expression is significantly upregulated in the medial preoptic area (MPA), paraventricular nucleus (PVN), and suprachiasmatic nucleus in the hypothalamus, and in the reticular thalamic nucleus, choroid plexus of lateral ventricules and tanycytes (Ta) of the ventral portion of the third ventricule[49]. Among these areas, the presentation of 2MT upregulated c-fos mRNA expression only in the MPA and PVN, but not in other areas in the mouse brain (Fig. 5). 2MT stimulation induces c-fos mRNA expression in the central nucleus of the amygdala[16]. However, c-fos mRNA expression in this area is not induced in the hibernating state in the squirrels[49]. These results indicate that although there are some similarities, clear differences exist for c-fos mRNA induction areas in the brain between 2MT stimulation and hibernating state. Thus, the hypothermic state induced by 2MT stimulation appears to be different from the hibernation state in terms of brain activity.

Since both 2MT stimulation and hibernation induce c-fos mRNA expression in the MPA and PVN, neural activity in these brain regions may induce hypothermia. If so, it is likely that these

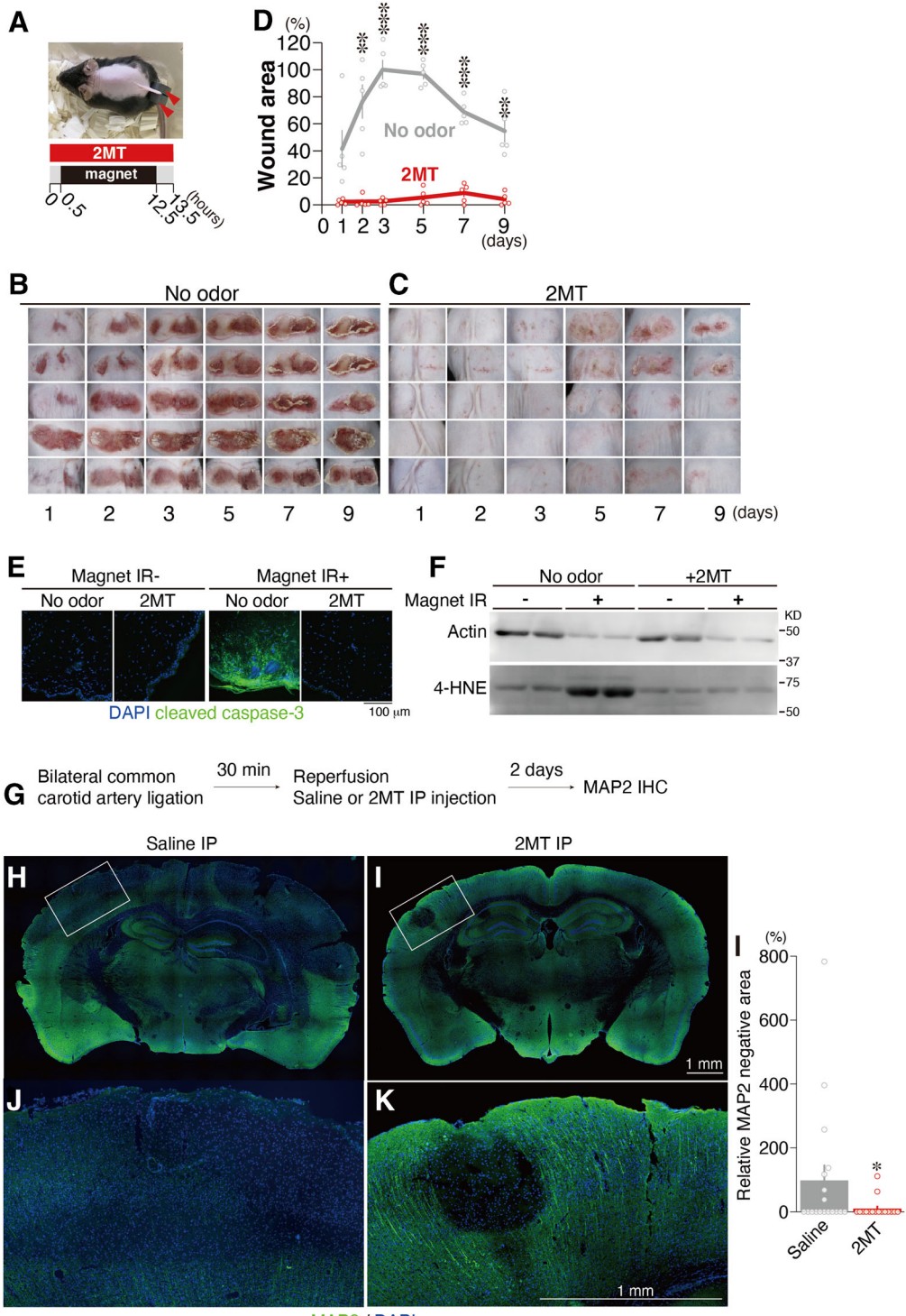

**Fig. 4 Suppression of ischemia/reperfusion injury by innate fear stimuli. A** Timeline of cutaneous ischemia/reperfusion experiments. **B–D** Photographs of cutaneous ischemia/reperfusion lesions for the no-odor control (**B**) and 2MT-treated (**C**) animals at each time point after reperfusion ($n = 5$ each). The percentage of lesioning at each time point after reperfusion with (red) and without (gray) 2MT stimulation relative to the wound area in the no-odor condition 3 days after reperfusion is also shown (**D**). **E** Representative images of immunohistochemistry for cleaved caspase-3 in the control (magnet IR$^-$) and ischemia/reperfusion (magnet IR$^+$) areas with and without 2MT administration. **F** Immunoblots of actin and 4-HNE in cutaneous lysates from control (magnet IR$^-$) and ischemia/reperfusion (magnet IR$^+$) areas. **G–L** Timeline of cerebral ischemia/reperfusion experiments (**G**), and representative images of immunohistochemistry for MAP2 in coronal brain sections from the saline and 2MT-treated animals (**H–K**). The areas indicated by white boxes in (**H** and **I**) are magnified and shown in (**J** and **K**). The mean percentages of MAP2-negative areas are also shown for both groups (**L**; $n = 18$ for the control group and $n = 16$ for 2MT-treated group). The size of the MAP2-negative area for the saline condition was set at 100%. Data are means ± SEM. Student's $t$ test or Mann–Whitney test was performed. *$P < 0.05$; **$p < 0.01$; ***$p < 0.001$.

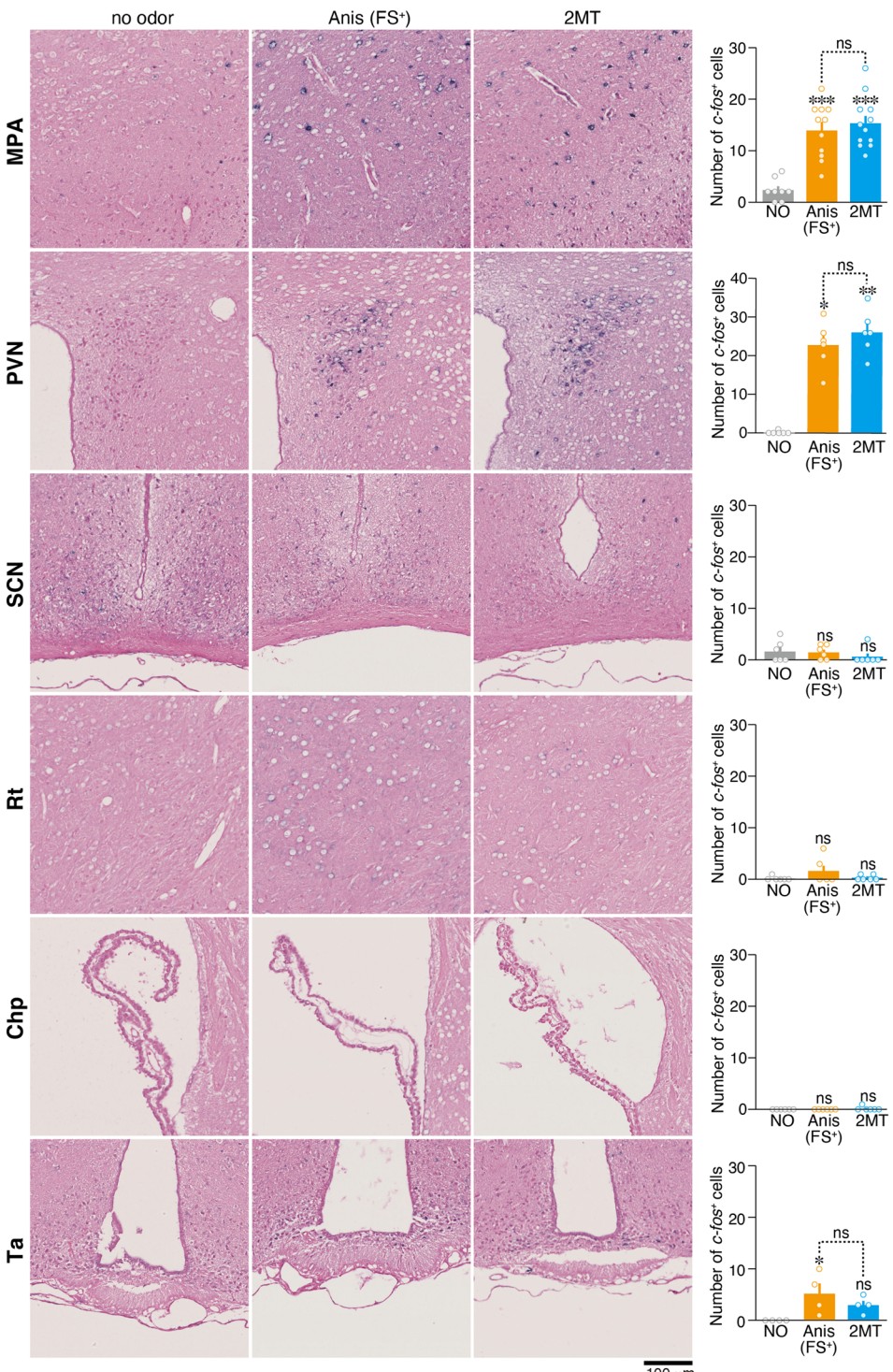

**Fig. 5 C-fos mRNA expression analysis of the hibernation/torpor brain areas.** Representative images of in situ hybridization of c-fos mRNA in response to no odor, Anis (FS+), and 2MT in the MPA [$n = 8$ for no odor, $n = 11$ for Anis (FS+), and $n = 12$ for 2MT], PVN ($n = 6$ each), SCN ($n = 6$ each), reticular thalamic nucleus (Rt) ($n = 6$ each), choroid plexus (Chp) ($n = 6$ each), and Ta ($n = 4$ each). Quantification of c-fos+ cells are also shown. Data are means ± SEM. One-way ANOVA followed by Tukey's multiple comparison was performed. *$P < 0.05$; **$p < 0.01$; ***$p < 0.001$; n.s., $p > 0.05$.

areas are not activated by learned fear odor, because it did not induce hypothermia (Fig. 1B, C). However, c-fos mRNA expression in the MPA and PVN was also upregulated in learned fear conditions (Fig. 5), suggesting that neural activities in the MPA and PVN do not only reflect the induction of hypothermia but are also related to common responses between innate and learned fear conditions.

Pacap-positive neurons in the MPA are activated in the torpor state in mice, and, furthermore, optogenetic and chemogenetic activation of these neurons induces a hibernation-like state[7–9]. 2MT stimulation may act as a sensory stimulus that can activate these neurons. However, the neuronal pathway that connects central brain activation and 2MT stimulation remains unclear.

We have previously shown that 2MT induces fear-related behaviors by activating TRPA1 in the trigeminal nerve, which transmits information to the Sp5 in the brainstem area[21]. Therefore, it is possible that information transmitted to the brainstem may play an important role with respect to 2MT-induced physiological responses. *Trpa1* is expressed not only in the trigeminal nerve but also in the vagus nerve. Correspondingly, 2MT stimulation induced the expression of *c-fos* mRNA in the Sp5, which receives axonal projection by the trigeminal nerve, as well as in the nucleus of the solitary tract (NST), which receives axonal projection from the vagus nerve (Supplementary Fig. 5). Therefore, 2MT stimulation may induce hypothermia and anti-hypoxia by activating neural pathways that originate from these brainstem areas. Therefore, in the present study, we aimed to elucidate the neural pathway originating from these brainstem areas.

**Central crisis pathway mediates tFO-induced life-protection.** TFOs activate the spinal trigeminal nucleus (Sp5) and NST in the brainstem via TRPA1 in the trigeminal and the vagus nerves (Supplementary Fig. 5A–J)[21]. Thus, we hypothesized that corresponding pathways originate from the NST/Sp5 and terminate in the region, which receive axonal projections from the NST/Sp5 and where the expression of the neural activity marker is induced by tFO stimulation. We searched for the candidate pathway using combination of the mouse brain connectivity atlas [(Allen Mouse Brain Connectivity Atlas (2011)] and our database of tFO-induced whole-brain activity mappings.

We assumed that the brain regions involved in the induction of crisis response are activated by odorants with crisis-response activities, but are not activated by odorants without these activities. In order to identify relevant brain regions, we analyzed hypoxia resistance induced by three odorants that share similar structures with 2MT. As a result, we found that in a 4% oxygen environment, 4-ethyl-2-methyl-2-thiazoline (4E2MT) conferred hypoxia resistance, whereas 2-methyl-2-oxazoline (2MO) and thiophene (TO) did not (Fig. 6C). Neurons in the NST/Sp5 send axonal projections into multiple areas in the brainstem, midbrain, and thalamus (Fig. 6A, B). Among these areas, 4E2MT induced expression of *c-fos* in the medullary reticular nucleus (MDRN) in the brainstem, and periaqueductal gray, superior colliculus, and parabrachial nuclei (PBN) in the midbrain (Fig. 6A, B).

The *c-fos* expression was upregulated in the periaqueductal gray and superior colliculus in response to 2MO, in addition to 2MT and 4E2MT, which have bioprotective activities (Fig. 6D, E), indicating that *c-fos* expression in these areas are not related to life-protection activities, but reflect sensory stimulations. Contrary to this, *c-fos* expression in medullary reticular nucleus and PBN was upregulated by the tFOs with bioprotective activities but not with non-bioprotective odorants (Fig. 6F, G); in particular, *c-fos* expression was markedly upregulated in the PBN. Furthermore, unlike brain regions where *c-fos* expression is induced by hibernation/torpor (Fig. 5), *c-fos* expression in the PBN was not induced by a learned fear odor presentation (Supplementary Fig. 6). These results raise the possibility that neural pathway from the Sp5/NST to the PBN might be responsible for life-protective effects induced by tFOs. To test this possibility, we analyzed whether virus-mediated chemogenetic activation of this pathway can orchestrate life-protective effects like tFO stimulation.

In the present study, we assessed the NST-PBN pathway using virus-mediated chemogenetic manipulation. We injected the retrograde-tracing AAVrg-Syn-Cre into the PBN, and injected Cre-dependent AAV-Syn-FLEX-hM3Dq-mcherry or AAV-Syn-FLEX-mCherry into the NST to express hM3Dq-mCherry or

mCherry in the NST-PBN pathway (Fig. 6H). In these animals, mCherry-labeled cells were detected mainly in the NST, indicating that the NST-PBN pathway can be specifically manipulated by chemogenetic activation (Fig. 6I–J and Supplementary Fig. 7A). Artificial activation of hM3Dq-positive cells by IP administration of clozapine-n-oxide (CNO) induced prominent hypothermia (Fig. 6K). Thereafter, we injected Cre-dependent AAV-Syn-FLEX-hM4Di-mcherry into the NST to inactivate NST-PBN pathway; however, no significant inhibitory effect was observed ($p = 0.33$, unpaired Student's $t$ test; Supplementary Fig. 8A, B). Since both the Sp5 and NST neurons project to the PBN, we hypothesized that the inhibition of the neural activity in the PBN might inhibit 2MT-induced hypothermia. Indeed, inhibition of the neural activity in the PBN by stereotaxic injection of muscimol, a GABA-A agonist, significantly inhibited 2MT-induced hypothermia ($p = 0.0041$, unpaired Student's $t$ test; Supplementary Fig. 8C–G). Artificial activation of hM3Dq-positive cells by IP administration of CNO induced prominent freezing behaviors (Fig. 6L). Kinetics and strength of these CNO-induced freezing behavior and hypothermia were similar to those induced by IP injection of tFOs (Supplementary Fig. 7C, D), indicating that chemogenetic activation of the NST-PBN pathway well mimics tFO-induced responses. Furthermore, chemogenic activation of the NST-PBN pathway suppressed oxygen consumption (Fig. 6M). These results indicate that sensory representation of tFOs activate central crisis pathway, which start from NST/Sp5 in the brain stem to PBN in the midbrain to orchestrate the crisis-response mode.

## Discussion

Innate fear has evolved to induce behavioral and physiological responses that increase the chance of survival in the natural world when animals are faced with life-threatening situations; however, the protective effects conferred by innate fear remain largely unknown. In the natural world, the animal brain perceives lethal dangers by integrating sensory inputs from multiple sources (e.g., injury due to biting, feeling of dyspnea caused by constriction, and odors/visuals/sounds emitted from predators). As it is thought to be difficult to mimic innate fear conditions perceived in the natural world in experimental paradigms, the precise range of responses among animals facing serious threats remains to be clarified. Importantly, tFOs including 2MT can induce more potent innate fear responses than ever via TRPA1 activation[16,21], likely because they act as supernormal stimuli that strongly activate fear perception systems in the brain. By artificially inducing innate fear in mice, we demonstrated that intrinsic life-protective abilities are conferred in crisis situations for the first time, shedding light on the latent relationship between innate fear and life preservation.

In this paper, we showed that the innate fear elicited by tFO presentation induces hypothermia and hypometabolism. In contrast, the presentation of a conditioned fear odor induces an increase in body temperature. Why are the opposite physiological responses induced depending on the type of fear stimulus? Physiological responses that increase body temperature help accelerate the dynamic flight or fight response. Many organisms are known to induce a state of suspended animation rather than escape when captured by a natural predator. A state of suspended animation induces a decrease in heart rate and body temperature[35]. Correspondingly, the present study shows that innate fear due to confinement-induced decreased heart rate and body temperature. In a severe crisis situation that cannot be responded with a flight or fight response, a physiological state based on hypothermia/hypometabolism may be induced. Indeed, the frequency of predatory attacks is reduced for animals in a state of suspended animation[50]. In addition to the implications in attack

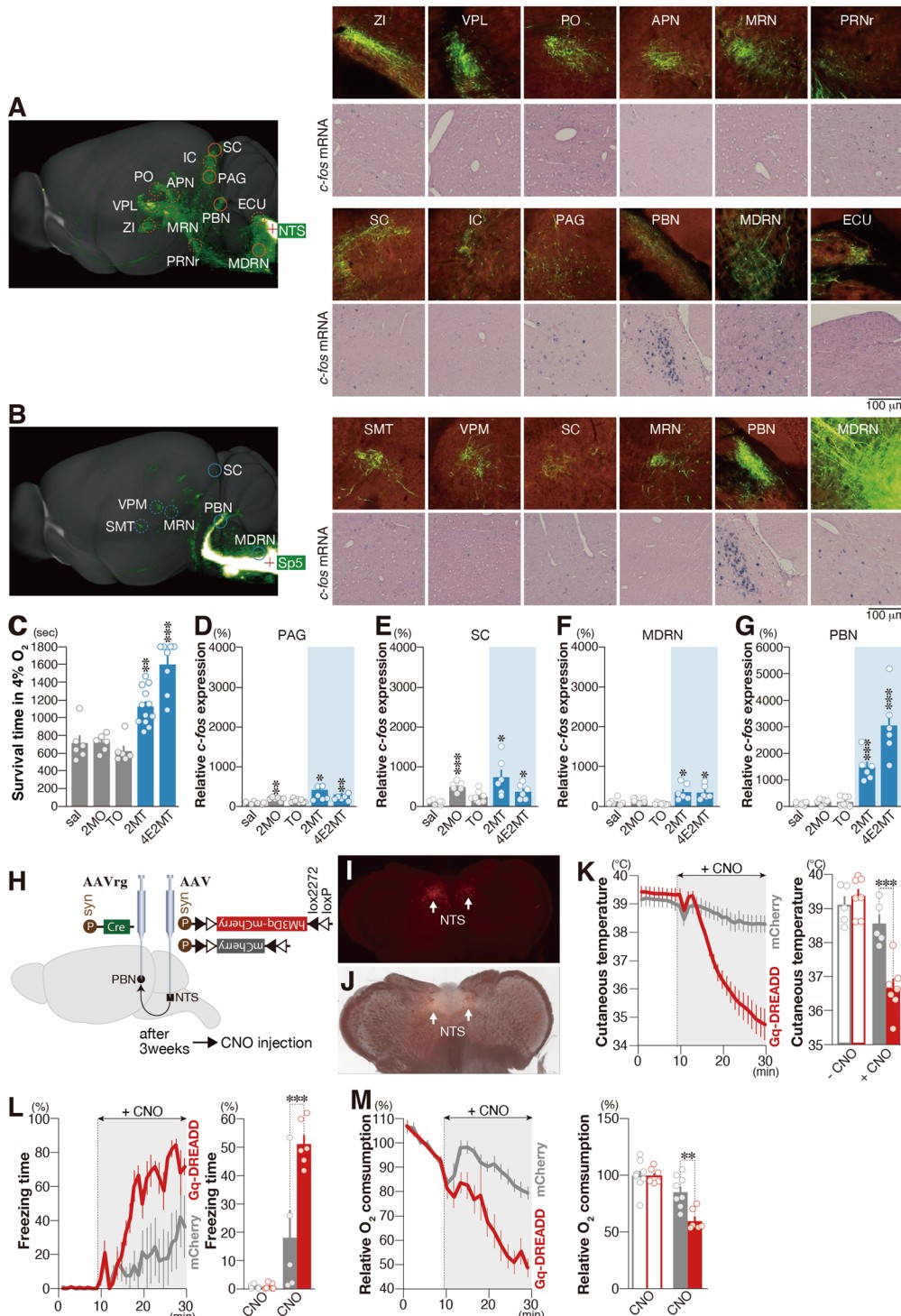

avoidance, the crisis-response mode elicited by innate fear stimuli may also induce protective effects against hypoxic encephalopathy triggered by respiratory depression due to capture by predators. Accordingly, we speculate that tFO stimuli strongly activate the innate fear system and induce survival state, a specific physiological state based on hypothermia/hypometabolism that is different from the flight or fight response.

The body temperature decreases to near-ambient temperatures in hibernating animals, which consequently confers resistance to I/R injury[51]. Accidental hypothermia (e.g., drowning in cold water) also exerts protective effects on the human brain[52]. Therapeutic hypothermia improves the neurological outcomes of

patients with cardiac arrest[4,53]. To protect neurons from ischemia-induced shortage of oxygen, inducing hypothermia is considered to be an effective strategy to reduce metabolism[54]. However, unlike hibernating animals, neurons in non-hibernating animals cannot survive in a cryogenic environment because of cytoskeleton disruption caused by excessive generation of reactive oxygen species[55]. Therefore, currently, the medical application of artificial hibernation technology requires a combination of technologies to protect cells in a low-temperature environment and efficiently lower body temperature and metabolism[56]. Although the body temperature of bears only lower by 4–5 °C during hibernation, their metabolism is suppressed much

**Fig. 6 Identification of the central crisis pathway. A**, **B** The mouse connectivity data for AAV-EYFP injection in the NST (**A**) and the Sp5 (**B**) derived from the Allen Mouse Brain Connectivity Atlas (2011) (sagittal images in the left and enlarged coronal images in the upper columns in the right panels), and representative images of in situ hybridization of *c-fos* mRNA in response to 4E2MT (enlarged coronal images in the lower columns in the right panels). **C** Mean survival time in 4% oxygen in response to IP administration of the odorants indicated ($n = 6$ for saline, $n = 6$ for 2MO, $n = 6$ for TO, $n = 12$ for 2MT, and $n = 8$ for 4E2MT). **D**–**G** Quantification of *c-fos* mRNA expression in response to IP administration of the indicated odorants in the PAG (**D**), SC (**E**), MDRN (**F**), and PBN (**G**) ($n = 6$ for saline, 2MO, 2MT, and 4E2MT, and $n = 8$ for TO). Mean expression in saline conditions was set at 100%. **H** Experimental design for chemogenetic activation of the NST-PBN pathway. **I**, **J** Representative fluorescent (**I**) and bright (**J**) images of hM3Dq-fused mCherry expression in the NST for AAV-FLEX-hM3Dq-mCherry injected animal. **K**–**M** Temporal analyses of cutaneous temperature (**K**), freezing behavior (**L**), and oxygen consumption (**M**) after CNO administration are shown for AAV-FLEX-hM3Dq-infected (red) and control (gray) mice (**K**; $n = 5$ for control and $n = 7$ for hM3Dq, **L**; $n = 5$ for control and $n = 6$ for hM3Dq, **M**; $n = 8$ for control and $n = 6$ for hM3Dq). Statistical significance was assessed for 20 min after CNO administration. Data are shown as mean ± SEM. Student's *t* test followed by Bonferroni correction or two-way ANOVA followed by Sidak's multiple comparisons was used to assess significance. *$P < 0.05$; **$p < 0.01$; ***$p < 0.001$. ZI zona incerta, VPL ventral posterolateral nucleus of the thalamus, PO posterior thalamic nuclear group, APN anterior pretectal nucleus, MRN midbrain reticular nucleus, PRNr pontine reticular nucleus, SC superior colliculus, IC inferior colliculus, PAG periaqueductal gray, PBN parabrachial nucleus, MDRN medullary reticular nucleus, ECU external cuneate nucleus, SMT submedial nucleus of the thalamus, VPM, ventromedial nucleus of the thalamus.

more than that expected from the temperature decrease, suggesting that body temperature and metabolism can be controlled by independent mechanisms[57]. IP injection of 2MT reduced body temperature by ~4 °C. Importantly, even under such conditions, hypoxia resistance and suppression of brain I/R injury were induced. Furthermore, IP injection of SBT reduced body temperature by only ~2 °C; however, SBT induced stronger anti-hypoxic activity than 2MT (Supplementary Fig. 5K–N). Thus, for tFO-induced crisis-response mode, anti-hypoxic activity can be controlled independently of body temperature suppression.

In the present study, the mechanisms of induction of hypoxia resistance ability independent of hypothermia have not been fully elucidated, although several implications were obtained. 2MT stimulation upregulated glucose uptake to accelerate glycolysis. This effect was not observed in the liver. Therefore, 2MT stimulation may induce a specialized metabolic state in the brain. Calcium levels inside neurons are maintained at lower concentrations than extracellular levels. Abnormally elevated calcium concentrations inside neurons can lead to protein denaturation and the induction of apoptosis. To prevent this, neurons continue to drive the calcium pump, which requires an energy source. Under hypoxic conditions, neurons fall into an energy-deficient state and require an alternative energy source. 2MT stimulation may prevent neuronal destruction by accelerating glycolysis, which produces energy without the need for oxygen. Continued driving of the electron transfer chain under oxygen-deficient conditions can lead to the generation of large amounts of ROS, resulting in cell destruction. 2MT stimulation may prevent the generation of ROS by suppressing the activity of PDH through phosphorylation, thereby cutting off the fuel supply to the electron transfer chain.

There are similarities between the hibernation state and innate fear-induced survival state, such as hypothermia and hypometabolism; however, their biological significances are clearly different. Hibernation aims to suppress the energy expenditure to survive in resource-limiting environments such as cold winter. In contrast, innate fear-induced survival state aims to orchestrate the intrinsic life-protective abilities to survive in crisis situations using energy as needed. For example, glucose uptake is almost completely suppressed during hibernation[45,58]; however, they are significantly upregulated in tFO-induced survival state. For therapeutic applications, especially in emergency care settings, we speculate that tFO-induced survival state predominates over the induction of therapeutic hypothermia, because it is directly linked to life-protective abilities in crisis situations.

In conclusion, our results demonstrated that the presentation of tFOs as a supernormal stimulus induced potent innate fear leading to an artificial hibernation state that is different from natural hibernation states. It also allowed for the survival of the mice in a state of lethal hypoxia. This state is regulated by sensory activation of the brainstem to midbrain pathway. In addition, tFO stimulation exerted therapeutic effects on cutaneous and brain I/R injury. The present study revealed a hidden relationship between tFO-induced innate fear and the latent intrinsic life-protective abilities. If preserved in humans, it is promising as a new medical technology.

## Methods

**Mice**. Male C57BL/6NCr mice were purchased from Japan SLC Inc. (Shizuoka, Japan). Ucp1$^{-/-}$ (stock number 17476) was purchased from The Jackson Laboratory (Bar Harbor, ME, USA), housed under a standard 12-h light/dark cycle, and allowed ad libitum access to food and water. Mice were 9–13 weeks old at the start of testing. The experimental protocols were approved by the Animal Research Committee of Kansai Medical University.

**Odorants**. We purchased 2MT, 4E2MT, 2MO, and TO from Tokyo Chemical Industry Co., Ltd. (Tokyo, Japan). Anis was purchased from Nacalai Tesque Inc. (Kyoto, Japan). TMT was purchased from Contech (Waterford, CT, USA). SBT was synthesized in-house as described previously[59]. Odor presentation was performed by introducing a piece of filter paper scented with 271 μmol of a test odorant into the test cage. Administration of odorant was performed by intraperitoneally injecting 100 μL of 1% odor solution. Fixed concentrations of odor gases were generated using calibration gas generation equipment (permeater PD-1B-2; Gastec, Kanagawa, Japan). When a calibration gas generation equipment was used, a constant concentration of gas was generated from test odorant and a sealed test cage ($31.5 \times 19.5 \times 13$ cm$^3$) was filled with the gas. In a test cage, a small door ($7.5 \times 8.5$ cm$^2$) was opened on the cage top to introduce a mouse, and a small ventilating hole (1 cm diameter) was made at the upper portion of cage wall. A nozzle placed at the lower portion of the front wall of a test cage was connected to the calibration gas generation equipment via a nafuron tube, and the gas was poured until the air in a test cage was completely replaced by odorous gas. Odor presentation was performed in a chemical fume hood to avoid cross-contamination.

**Olfactory fear conditioning**. Olfactory fear conditioning was performed according to a method[16]. Briefly, C57BL/6 mice were placed into the conditioning chamber and allowed 3 min of free exploration; then, one of two odorants (Anis or eugenol) was introduced into the conditioning chamber for 30 s. An FS was only delivered when Anis was introduced into the chamber. Mice were exposed to Anis-FS and eugenol-no FS a total of six times each in a randomized order, with an inter-trial interval of 4 min. All stimuli were presented and sequenced under the control of dedicated software (Freeze-Frame2; Actimetrics, Wilmette, IL, USA). Olfactory fear conditioning was performed 1 day prior to testing for all experiments.

**Measurement of cutaneous temperature**. C57BL/6 mice were anesthetized with pentobarbital (50 mg/kg, IP) 2–3 days prior to testing, and the fur on the back was removed with a chemical hair remover.

To analyze the effects of innate and learned fear stimuli on cutaneous temperature, each mouse was placed in a separate test cage ($17.5 \times 10.5 \times 15$ cm$^3$), habituated for 10 min, and subjected to consecutive odor presentations: (1) no-odor for 10 min, (2) eugenol for 10 min, and (3) test odorant (2MT or Anis-FS+) for 20 min. Cutaneous temperature was recorded using an infrared digital thermographic camera (Infrared Thermography H2640; NEC Avio Infrared

Technologies Co., Ltd., Tokyo, Japan) at 10 frames/s. Cutaneous temperature on the back was automatically analyzed ($n \geq 8$ each) using specially designed software[60]. The change in cutaneous temperature was calculated as the difference in temperature between no-odor presentation (10 min) and test odorant presentation (20 min). Control values were set to 0.

To analyze the effects of IP injection of odorants, mice were habituated for 10 min. Following habituation, cutaneous temperature was recorded in the no-odor condition for 10 min, and 100 μL saline or 1% odorant solutions (2MT, TMT, or SBT) were intraperitoneally injected. Cutaneous temperature was recorded for 30 min following odorant injection. Cutaneous temperature on the back was automatically analyzed ($n = 4$ for saline, TMT, and SBT; $n = 3$ for 2MT; $n = 6$ for 4E2MT) as described above. During the recording of cutaneous temperatures, freezing behavior was also recorded and analyzed using a video-based measurement system (Freeze-Frame2, ActiMetrics, Evanston, IL, USA)[16]. The mice were considered to freeze if the movement was not detected for 2 s.

**Measurement of core body temperature and heart rate induced by innate and learned fear odors.** Core body temperature and heart rate were analyzed in freely moving animals using a surgically implanted radio-telemetry device. C57BL/6 mice were anesthetized with pentobarbital (50 mg/kg, IP) and implanted with a radio-telemetry transmitter (TA11ETA-F10; DataSciences International, St. Paul, MN, USA) in accordance with the surgical procedure described by the manufacturer. Briefly, the abdomen was opened, and the transmitter was placed into the peritoneal cavity with the leads located at the right shoulder (negative lead) and left chest (positive lead). The transmitter was sutured to the muscle layer and the muscle and skin were closed in layers. After the surgery, the mice were allowed to recover for ~10 days before testing. On the test day, each mouse was placed in a separate test cage ($17.5 \times 10.5 \times 15\ cm^3$), habituated for 10 min, and subjected to three consecutive odor presentations: (1) no-odor for 10 min, (2) eugenol for 10 min, and (3) test odorant for 20 min. Physiological parameters were automatically transmitted from the device every 10 s using Dataquest A.R.T. software (DataSciences International; $n \geq 6$ each). Changes in heart rate and core body temperature were calculated as the differences in heart rate and core body temperature between no-odor presentation and test odorant presentation (10 min duration for heart rate and 20 min duration for core body temperature), respectively.

For the analysis of long exposure of 2MT, core body temperature was measured in the absence of an odorant for 10 min and subsequently during test odorant presentation (no-odor control and 2MT) for 12 h ($n = 6$ each) by presenting four sets of filter papers dropped with 25 μL of odorants in the sample cage (Innocage® Mouse; Innovive, San Diego, CA, USA). For analysis of locomotor activities, mouse behavior was recorded by a digital video recorder (HDR-PJ790V, Sony) and recorded files were converted to a decompressed AVI file by VitualDub. Locations of a mouse were measured using particle analysis in ImageJ 1.50i. The data were loaded into R software V3.6.1, then distance traveled and trajectory were analyzed using R package trajr[61] at a rate of one frame per second. One week after the odorant exposure, mice were transferred to a test cage ($28 \times 18 \times 13.5\ cm^3$). The distance traveled and trajectory over 10 min were automatically analyzed using a LimeLight2 software (ActiMetrics).

**LiCl conditioning.** For LiCl conditioning, a radio-telemetry transmitter was implanted on the right side of the peritoneal cavity of C57BL/6 mice to allow injection into the center of the peritoneal cavity. After a 10-day recovery period, LiCl conditioning was performed according to a method[62], with minor modifications. On experimental day 1, a piece of filter paper ($2 \times 2\ cm^2$) with 271 μmol of Anis was placed into a training cage ($28 \times 18 \times 13.5\ cm^3$), following which a single mouse was introduced into the cage. After 10 min of odor presentation, the mouse was injected with LiCl (127 mg/kg) or saline (control) and returned to the training cage for an additional 30 min. During this additional period, pieces of filter paper scented with 271 μmol were added three times at 10-min intervals. This training procedure was repeated once per day for 7 days. On day 8, conditioned responses were analyzed as follows: each mouse was placed in a separate test cage ($17.5 \times 10.5 \times 15\ cm^3$), habituated for 10 min, and subjected to three consecutive odor presentations: (1) no-odor for 10 min, (2) eugenol for 10 min, and (3) Anis for 20 min. Physiological responses (heart rate and core body temperature) were analyzed during these consecutive odor presentations ($n = 6$ each). Changes in core body temperature and heart rate were calculated as the differences in body temperature and heart rate between no-odor presentation and test odorant presentation (10 min), respectively. Saline control values were set to 0.

The control experiment, in which physiological responses to LiCl injection were measured, was performed as follows: after recovery from the telemetry probe transplantation procedure, each mouse was placed in a separate test cage ($17.5 \times 10.5 \times 15\ cm^3$), allowed to habituate for 10 min, and subjected to a no-odor control for 10 min to collect baseline measurements. Immediately after the baseline measurement, saline or LiCl was intraperitoneally injected, and physiological responses were measured ($n = 6$ each). Changes in core body temperature and heart rate were calculated as the differences in body temperature and heart rate between no-odor presentation and post injection (10 min). Saline control values were set to 0.

**Restraint experiment.** C57BL/6 mice were implanted with radio-telemetry probes 10 days prior to the experiment, as described above. On test day, each mouse was placed in a separate test cage ($17.5 \times 10.5 \times 15\ cm^3$), allowed to habituate for 10 min, and physiological parameters (heart rate and core body temperature) were analyzed for 10 min. Immediately afterwards, mice were restrained in ventilated 50-mL plastic tubes (Becton, Dickinson and Company, Franklin Lakes, NJ, USA), and physiological parameters were analyzed for 20 min ($n = 6$ each). Changes in core body temperature and heart rate were calculated as the differences in core body temperature and heart rate between 10 min before restraint and 5 min after restraint. Control (no restraint) values were set to 0.

**Cutaneous blood flow.** Cutaneous blood flow was analyzed using a laser Doppler blood flow monitor (ALF21D; Advance Company Ltd, Tokyo, Japan). Five days prior to the test day, C57BL/6 mice were anesthetized with pentobarbital (50 mg/kg, IP) and the fur on the back was removed with a chemical hair remover. On the next day, mice were restrained in ventilated 50-mL plastic tubes (Becton, Dickinson and Company) and allowed to habituate for 40 min. In addition to ventilation holes, a small window ($1.5 \times 2.5\ cm^2$) was opened in the tube to allow access to the skin on the back. This habituation procedure was repeated once per day for 4 days. On the test day, mice were restrained in the tube, and a laser Doppler probe was affixed to the back using adhesive tape. After stable blood flow signals were obtained during the habituation period (~30 min), mice were subjected to three consecutive odor presentations: (1) no-odor for 10 min, (2) eugenol for 10 min, and (3) test odorant (271 μmol 2MT [unpaired] or 271 μmol Anis that had been previously paired with FS for 20 min) ($n \geq 8$ each). Each odorant was presented to the mouse's nose using a filter paper. Blood flow signals were automatically extracted every 10 s using the included software. Mean blood flow values during no-odor presentation (10 min) and test odorant presentation (20 min) were calculated. Control (no-odor) values were set to 100%.

**Core body temperature in the thermoneutral condition and in Ucp1-KO mice.** C57BL/6 mice were implanted with radio-telemetry probes and allowed to recover for 10 days in individual home cages at ambient temperature (~25 °C).

For the thermoneutral experiment, we adapted a previously described thermoneutral housing method[63]. Following convalescence, C57BL/6 mice were housed in incubators set to 30 °C or at ambient temperature ($n = 5$) for 10 days. On test day, physiological responses were analyzed in an incubator set to 30 °C. Mice were immediately transferred into the 30 °C test cage, and core body temperature and heart rate were measured in the absence of an odorant for 30 min, following which 2MT was presented for 30 min. Mean core body temperatures were calculated during no-odor presentation (30 min) and 2MT presentation (30 min).

Ucp1-KO mice ($n = 8$) and littermate controls ($n = 7$) were transferred to the test cage, and core body temperature was measured in the absence of an odorant for 10 min, following which 2MT was presented for 30 min. Mean core body temperature was calculated during no-odor presentation (10 min) and 2MT presentation (20 min).

**Measurements of respiratory rate and oxygen saturation.** Respiratory rate and oxygen saturation were analyzed in C57BL/6 mice using a pulse oximeter (MouseOx® Plus; STARR Life Sciences Corp., Oakmont, PA, USA). One week prior to test day, an oximeter probe was attached to the neck of each mouse and mice were allowed to habituate for 60 min. This habituation procedure was repeated once per day for 5 days. Two days prior to the test day, mice were anesthetized with pentobarbital (50 mg/kg, IP), and the fur around the neck was removed with a chemical hair remover. On the test day, mice were affixed with oximeter probes, placed into sampling chambers, and habituated until signals stabilized (~60 min). After the habituation period, baseline values were collected in the absence of an odorant for 10 min. Immediately afterwards, a piece of filter paper scented with 271 μmol of 2MT or eugenol was placed into the sampling chamber and oximeter signals were analyzed for an additional 20 min ($n = 6$). Oximeter signals were automatically collected at 1 Hz using the included software. Mean respiratory rates and oxygen saturation were calculated during no-odor presentation (10 min) and test odorant presentation (20 min).

**Measurement of oxygen consumption.** Oxygen consumption was analyzed every 1 min using a mass spectrometric calorimeter (Arco-2000; Arco System, Chiba, Japan). C57BL/6 mice were introduced into a test chamber and habituated for >60 min. Following habituation, oxygen consumption was measured in the absence of an odorant for 10 min, and ten pieces of filter paper scented with 271 μmol of 2MT or saline were placed into the test chamber, following which oxygen consumption was analyzed for an additional 30 min ($n = 8$). Mean oxygen consumption values were calculated during the no-odor presentation (10 min before test odorant presentation) and test odorant presentation (20 min after test odorant presentation).

**Hypoxia resistance.** To analyze the effects of 2MT presentation, C57BL/6 mice were introduced into separate cages (Innocage® Mouse) and exposed to a total of 100 μL of 2MT (four pieces of $2 \times 2\ cm^2$ filter paper each spotted with 25 μL of 2MT). The cage was then covered with a lid for either 10 min or 50 min. Following

exposure to 2MT, mice were moved to separate test chambers ($17 \times 17 \times 18.5$ cm$^3$). Each chamber had two holes on opposite sides at different heights (4.5 and 12 cm, respectively) and a wire mesh platform (height: 9 cm) where mice were confined for the duration of the experiment. To produce an environment with 4% O$_2$, compressed nitrogen gas and compressed air cylinders were connected to two gas permeater (PD-1B-2; Gastec Corp., Kanagawa, Japan). A mixture of 1600 mL/min nitrogen gas and 400 mL/min air was poured into the test chamber through the upper hole. Elapsed time was measured between the closing of the chamber and the last breath of each mouse ($n = 6$ each).

To analyze the effects of restraint, C57BL/6 mice were restrained in ventilated 50-mL plastic tubes for either 10 or 30 min and survival times under 4% oxygen were analyzed as described above.

To analyze the effects of learned fear stimuli, C57BL/6 mice were conditioned by pairing Anis with electric shocks as described above. On the next day, the mice were introduced into separate cages (Innocage® Mouse) and exposed to a total of 100 μL of Anis (four pieces filter paper each spotted with 25 μL of Anis). The cage was then covered with a lid for 50 min, then survival times under 4% oxygen were analyzed as described above ($n = 6$). The survival times were also analyzed for nonconditioned control mice ($n = 6$).

To determine the effective concentration of 2MT odor gas, C57BL/6 mice were introduced into a cage, which was filled with a fixed concentration of 2MT gas using gas permeater. After 30 min of odor presentation, survival time under 4% oxygen was analyzed ($n = 6$ each). For the control experiment, C57/BL6 mice were introduced into a cage without 2MT odor, and survival time under 4% oxygen was analyzed ($n = 36$).

To analyze the effects of corticosterone, C57BL/6 mice were IP injected with saline or corticosterone[16,64,65] (2 mg/kg; Sigma-Aldrich). At 30 min after drug administration, survival times under 4% oxygen were analyzed as described above.

To analyze the effects of IP injection of odorants, C57/BL6 mice were IP injected with 2MT, TMT, or SBT (40 mg/kg each) for Figure S4N, and were IP injected with 2MO, 2MT, 4E2MT (80 mg/kg each), or TO (40 mg/kg) for Fig. 6C. At 30 min after IP administration, survival times under 4% oxygen were analyzed as described above.

**GC-MS analysis.** C57BL/6 mice were exposed to 10 p.p.m. of 2MT gas for 30 min using calibration gas permeater, as described above. After odor presentation, blood samples were collected via decapitation and serum were prepared by centrifugation. Serum samples from three mice were mixed, and 2MT concentration was analyzed by Shimadzu Techno-Research Inc. Briefly, standard 2MT solutions and serum sample were extracted with methanol and added to 4 mL of saturated saline. For sample preconcentration, an SPME fiber (DVB/CAR/PDMS; Shimadzu, Kyoto, Japan) was applied, and autosampler (AOC-5000Plus; Shimadzu, Kyoto, Japan) was used for automatic adsorption and injection. Adsorption time was 20 min at 40 °C, and the fiber was withdrawn and transferred into the injection port of GC. Desorption time was 4 min, while the temperature of the injection port was set at 270 °C. GC-MS analysis was performed with a GC-MS-QP2010Ultra (Shimadzu, Kyoto). The column used was an inertCap Pure WAX (GL Sciences, Tokyo, Japan). The GC program started at 60 °C for 2 min, and was raised to 250 °C at a rate of 20 °C/min and held for 2 min.

**Live-cell metabolic assay.** Oxygen consumption rate in HepG2 cell in response to 2MT was analyzed by the XF Cell Mito Stress Test™ using a Seahorse XFp analyzer (Seahorse Bioscience, North Billerica, MA) according to the manufacturer's instruction[66].

**Metabolite extraction from mouse tissue.** C57BL/6 mice were introduced into separate cages (Innocage® Mouse), habituated for 2 h, and exposed to a total of 100 μL of 2MT or saline (four pieces of $2 \times 2$ cm$^2$ filter paper each spotted with 25 μL of 2MT or saline; $n = 6$ each), following which the cage was covered with a lid for 5 min. At 5 min of odor presentation, $^{13}$C$_6$-Glucose (558 mg/kg; Taiyo Nippon Sanso, Tokyo, Japan) was intraperitoneally injected and another 100 μL of 2MT or saline was introduced into the cage, which was again covered with a lid for 15 min. Following odor presentation, mice were decapitated and their brains and livers dissected and stored at −80 °C until extraction. The frozen samples (~50 mg) were completely homogenized using a cell disrupter (Shake Master NEO; Bio Medical Science, Tokyo, Japan) at 1500 r.p.m. for 2 min after the addition of 500 μL of methanol containing internal standards [20 μM each of methionine sulfone, 2-(N-morpholino)-ethanesulfonic acid and D-camphor-10-sulfonic acid]. The homogenate was then mixed with Milli-Q water and chloroform in a volume ratio of 5:2:5 and centrifuged at $4600 \times g$ for 15 min at 4 °C. Subsequently, 300 μL of the aqueous solution was centrifugally filtered through a 5-kDa cut-off filter (Human Metabolome Technologies, Tsuruoka, Japan) to remove proteins. The filtrate was centrifugally concentrated at $9100 \times g$ for 2 h at 20 °C. and dissolved in 50 μL of Milli-Q water that contained reference compounds (200 μM each of 3-aminopyrrolidine and trimesate) immediately prior to metabolome and glucose analysis.

**Metabolome analysis.** Metabolome analysis was performed using an Agilent G1600 CE capillary electrophoresis system, an Agilent 6210 time-of-flight mass

spectrometry (TOFMS), an Agilent1100 series isocratic HPLC pump, a G1603A Agilent CE-MS (capillary electrophoresis-MS) Adapter Kit and a G1607A Agilent CE-ESI (electrospray ionization)-MS (Sprayer Kit (all Agilent Technologies, Santa Clara, CA). The CE-MS Adapter Kit includes a capillary cassette that facilitates thermostating of the capillary, and the CE-ESI-MS Sprayer Kit that simplifies coupling the CE System with MS Systems was equipped with an electrospray source.

To analyze cationic compounds, a fused silica capillary (50 μm i.d. × 100 cm) was used with 1 M formic acid as the electrolyte[67-69]. Samples were injected with a pressure injection of 50 mbar for 5 s (~5 nL). The applied voltage was set at 30 kV, the capillary temperature was thermostated to 20 °C and the sample tray was cooled below 5 °C. The methanol/water (50% v/v) containing 0.1 μM hexakis(2,2-difluoroethoxy)phosphazene was delivered as the sheath liquid at 10 μL/min. TOFMS was performed in positive ion mode, and the capillary voltage was set to 4 kV. The fragmentor, skimmer, and Oct RFV (octapole radio frequency voltage) voltage was set at 75, 50, and 125 V, respectively. A flow rate of drying nitrogen gas (heater temperature 300 °C) was maintained at 7 L/min. Automatic recalibration of each acquired spectrum was achieved using the masses of the reference standards ([$^{13}$C isotopic ion of a protonated methanol dimer (2 MeOH+H)]$^+$, $m/z$ 66.0632) and ([hexakis(2,2-difluoroethoxy)phosphazene + H]$^+$, $m/z$ 622.0290).

To analyze anionic metabolites, a commercially available COSMO(+) (chemically coated with cationic polymer) capillary (50 μm i.d. × 105 cm) (Nacalai Tesque, Kyoto, Japan) was used with a 50 mM ammonium acetate solution (pH 8.5) as the electrolyte[69-71]. Samples were injected with a pressure injection of 50 mbar for 30 s (~30 nL) and the applied voltage was set at −30 kV. Methanol/5 mM ammonium acetate (50% v/v) containing 0.1 μM hexakis(2,2-difluoroethoxy) phosphazene was delivered as the sheath liquid at 10 μL/min. ESI-TOF-MS was performed in negative ion mode, and the capillary voltage was set to 3.5 kV. The fragmentor, skimmer, and Oct RFV voltage were set at 100, 50, and 200 V, respectively. Automatic recalibration of each acquired spectrum was performed using reference masses of reference standards ([$^{13}$C isotopic ion of deprotonated acetic acid dimer (2CH3COOH-H)]-, $m/z$ 120.03841) and ([hexakis + deprotonated acetic acid (CH3COOH-H)]-, $m/z$ 680.03554). The other conditions were identical to the cationic metabolite method.

**Glucose analysis.** Glucose analysis was performed using an Agilent G1600 CE capillary coupled with and an Agilent 6410 Triple Quad triple-quadrupole tandem mass spectrometer. A fused silica capillary (50 μm i.d. × 100 cm) was used with 300 mM diethylamine as the electrolyte. Samples were injected with a pressure injection of 50 mbar for 9 s (~9 nL). The applied voltage was set at 20 kV. Methanol/water (50% v/v) containing 0.2% diethylamine was delivered as the sheath liquid at 10 μL/min. The mass spectrometer was operated in the multiple reaction monitoring mode using the positive ionization and 3500 V of ion spray voltage was applied. The flow rate of nebulizer nitrogen gas and drying nitrogen gas (heater temperature 300 °C) was maintained at 10 psig and 10 L/min, respectively.

The Q1 (protonated precursor ion), Q3 (production), fragmentor, and collision energy for glucose-$^{12}$C were 179$m/z$, 89 $m/z$, 60 V, and 0 V, respectively.

**PDH assay.** C57BL/6 mice were introduced into separate cages (Innocage® Mouse), habituated for 2 h, and exposed to a total of 100 μL of 2MT or saline (four pieces of $2 \times 2$ cm$^2$ filter paper each spotted with 25 μL of 2MT or saline; $n = 6$ each), following which the cage was covered with a lid for 20 min. Following odor presentation, mice were decapitated and their brains dissected. Levels of total PDH, pPDH (S232), and pPDH (S300) expression and PDH activity in the brain lysates were measured using PDH ELISA Kits, in accordance with the manufacturer's instructions (Abcam, Cambridge, UK).

**Cutaneous I/R injury.** The cutaneous I/R injury model was adapted from a previously described method[72]. Briefly, mice were anesthetized with pentobarbital (50 mg/kg, IP) 2–3 days prior to testing and the fur on the back was removed with a chemical hair remover. Mice were introduced into separate cages with food, water, and bedding (Innocage® Mouse), exposed to a total of 100 μL of 2MT or water (four pieces of $2 \times 2$ cm$^2$ filter paper each spotted with 25 μL of 2MT or water), and the cage was covered with a lid for 30 min. After this 30-min period, mice were removed, the dorsal skin was gently pulled and trapped between two round ferrite magnetic plates (NeoMag Co., Chiba, Japan), and mice were returned to their cages. At this point, another set of four filter papers (scented with 2MT or water) was introduced into the cage, which was then covered with a lid for 12 h. After this 12-h period, the magnets were removed, and the wound area was photographed on a daily basis. Wound area images were converted to greyscale and signal intensities were quantified using Adobe Photoshop by a single-blinded investigator ($n = 5$ each). The signal intensities of the wound area in control mice at 3 days after reperfusion were set to 100%.

To analyze the expression of cleaved caspase-3, mice were perfused with 4% paraformaldehyde (PFA) 7 h after reperfusion and cutaneous samples including I/R areas were dissected. Samples were soaked in 30% sucrose/PBS overnight, following which they were embedded in OCT compound. Frozen sections with a thickness of 16 μm were incubated in blocking buffer (5% goat serum/0.3% Triton X-100/PBS) for 30 min at room temperature, following which they were incubated with anti-

cleaved caspase-3 antibody (1:500, Cell Signaling Technology, Danvers, MA, USA) in blocking buffer overnight at 4 °C. After the slides were washed in PBS, they were incubated with anti-rabbit Alexa-488 (1:1000, Invitrogen, Carlsbad, CA, USA) for 1 h at room temperature. Slides were covered with DAPI (4′,6-diamidino-2-phenylindole)-containing mounting medium (Vector Laboratories Inc., Burlingame, CA, USA), and fluorescent images were obtained using a DMI6000 B microscope (Leica Camera AG, Wetzlar, Germany).

To analyze the expression of actin and 4-HNE, cutaneous samples were dissected from areas with and without magnet I/R and total lysates were prepared in RIPA (radioimmunoprecipitation assay) buffer (50 mM Tris-HCl (pH 8.0), 150 mM NaCl, 1 mM EDTA, 1% NP-40, 0.5% sodium deoxycholate, 0.1% sodium dodecyl sulfate) with protease (Sigma-Aldrich) and protein phosphatase (Abcam) inhibitor cocktails. The protein concentration was measured using a BCA Protein Assay Kit (Thermo Fisher Scientific). Cell lysates were separated on Bolt 4–12% Bis-Tris Plus (Thermo Fisher Scientific) and transferred to PVDF membranes with iBlot 2 (Thermo Fisher Scientific). The membranes were incubated in a blocking buffer (5% skim milk/0.05% Tween-20/TBS) for 30 min at room temperature, then in a blocking buffer containing anti-4-HNE (1:1000, Abcam) overnight, followed by incubation with anti-rabbit conjugated with horseradish peroxidase (1:1000, Abcam) for 2 h at room temperature. Horseradish peroxidase signals were revealed with Chemi-Lumi One L (Nacalai Tesque Inc.). For actin detection, the membrane was washed with stripping buffer (62.5 mM Tris-HCl (pH 6.0), 2% SDS, 0.7% 2-mercaptoethanol) to strip out the anti-4-HNE antibody for 30 min at 50 °C, following which it was incubated with anti-actin (1:5000, Abcam) overnight at 4 °C. The signals were again revealed with Chemi-Lumi One L.

**Bilateral common carotid artery occlusion**. Following anesthetization with pentobarbital (50 mg/kg, IP), mice were placed on their backs on a heating pad. A midline incision was made at the neck and the common carotid artery was revealed via blunt dissection. After careful separation of the common carotid artery and vagus nerve on both the left and right sides, the common carotid arteries were clipped using micro-serrefines (18055-03, Fine Science Tools Inc., Foster City, CA, USA) for 30 min. A total of 100 μl of saline or 1% 2MT was intraperitoneally injected just prior to removing the clips. After suturing the wounds, animals were maintained in separate cages. For the sham procedure, the common carotid arteries were revealed and then left for 30 min without clipping. Animals were sacrificed and perfused with 4% PFA 2 days after the surgery. Brains were removed and soaked in 30% sucrose/PBS for several nights and embedded in OCT compound. Frozen sections with a thickness of 20 μm were incubated in blocking buffer (5% goat serum/0.3% Triton X-100/PBS) for 30 min at room temperature, following which they were incubated with anti-MAP-2 antibody (AB5543, 1:500, Merck Millipore, Burlington, MA, USA) in blocking buffer overnight at 4 °C. After washing in PBS, the slides were incubated with anti-chicken Alexa-488 (1:800, Jackson ImmunoResearch Laboratories Inc., West Grove, PA, USA) for 1.5 h at room temperature. Slides were covered with DAPI-containing mounting medium (Vector Laboratories), and fluorescent images were obtained using a DMI 6000B microscope (Leica Camera AG).

**Immediate-early gene mapping**. Immediate-early gene mapping was performed in accordance with a previously described method, with minor modifications[16]. Briefly, C57BL/6 mice were introduced into separate cages and habituated for 2 h. Following habituation, a filter paper scented with 271 mmol 2MT or Anis previously paired electric shocks was presented every 5 min for a 30-min period. Alternatively, mice were intraperitoneally injected with 100 μL of 1% 2MT, 2MO, TO, or 4E2MT. Following 30 min of odor presentation or odor injection, mice were anesthetized with gaseous isoflurane (Mylan, Canonsburg, PA) and perfused with ice-cold 4% PFA in phosphate-buffered saline (PBS). The brains were then removed and immersed in 4% PFA in PBS overnight at 4 °C. The fixed brains were dehydrated in a graded ethanol and xylene series and then embedded in paraffin using an automated system (Sakura rotary, RH-12DM; Sakura Finetek, Tokyo, Japan). Coronal sections with a thickness of 5 μm were prepared using an Automatic Slide Preparation System (AS-200S, Kurabo, Osaka, Japan). In situ hybridization was performed using an automated system (Discovery XT, Ventana Medical Systems, Oro Valley, AZ) according to the manufacture's protocol. To prepare antisense RNA probes for c-fos, DNA fragments spanning the 129–537 and the 543–1152 bp regions of c-fos were amplified by polymerase chain reaction from brain complementary DNA of C57BL/6 mice. The DIG-labeled probes (1:1000 dilution) were hybridized for 3 h using a RiboMap Kit (Roche) at 74 °C. The slides were then incubated with biotin-conjugated anti-DIG antibody (1:500, Jackson ImmunoResearch, West Grove, PA) at 37 °C for 28 min. The probe was detected using the Ventana BlueMap Kit (Roche, Basel, Switzerland) at 37 °C for 6 h, and counterstained with a Red Counterstain Kit (Roche) at 37 °C for 4 min. Coverslips were applied using an Automated System (Tissue Tek® GlasTM; Sakura Finetek). The stained images were scanned using a NanoZoomer virtual microscope system (2.0 RS, Hamamatsu Photonics, Hamamatsu, Japan). The number of c-fos+ cells in the stained images was then counted by single-blinded investigators. For the analysis of c-fos mRNA expression in the NST/Sp5-projecting areas in 4E2MT-treated animals, the stained images were converted to gray scale, and signal intensities were quantified using Adobe Photoshop (n = 6–8 for each region).

**Chemogenetic manipulation of NST-PBN pathway**. For virus injections into NST, C57BL/6 mice were anesthetized (50 mg/kg of pentobarbital, IP) and placed on a stereotaxic device with the head bent downward. An incision was made in the skin of the dorsal neck and muscles were dissected to reveal the membrane overlying the dorsal medulla. An AAV virus carrying a double-floxed inverted hM3D(Gq) fused with mCherry gene (AAV5-hSyn1-DIO-hM3D(Gq)-mCherry, Addgene), double-floxed inverted hM4Di(Gi) fused with mCherry gene (AAV5-hSyn1-DIO-hM4Di(Gi)-mCherry, Addgene) or mCherry gene (AAV5-hSyn1-DIO-mCherry, Addgene) was infused into the NST with 80 nL of virus solution at the following coordinates: anterior–posterior (AP), 0.4 mm caudal from the caudal end of cerebellum; left–right (LR), ±0.2 mm; dorsoventral (DV), 0.6 mm from the surface. For PBN injections, anesthetized mice were placed on a stereotaxic device and small hole were drilled on the skull. A retrograde AAV virus carrying a Cre gene (AAVrg-hSyn1-Cre, Addgene; 200 nL volume) was injected into the PBN (coordinates: AP, −5.1 mm; LR, ±1.7 mm; DV, −3.7 mm from bregma). Injections were performed bilaterally at a rate of 1 nL/s using a glass pipette connected to a Nanoject III (Drummond Scientific, Broomall, PA). More than 3 weeks after the viral injections, physiological responses of the animals were analyzed with an IP administration of 1 mg/kg of CNO (Sigma-Aldrich) as described above with some modifications; cutaneous temperature was measured from 10 min prior and to 30 min after a CNO injection; for oxygen consumption measurements, the mice were introduced into a test chamber and CNO solution was injected after 25 min of habituation. Oxygen consumption was measured from 10 min prior and to 30 min after the CNO injection. Brain was carefully dissected and fixed in 4% PFA solution. The brain was sectioned with a vibratome (LinearSlicer Pro10, D.S.K.) with a thickness of 100 μm and expression of mCherry was analyzed at the level of medulla using a fluorescence microscope (BZ-9000, Keyence).

**Muscimol infusion**. Mice were anesthetized with IP injection of pentobarbital (50 mg/kg) and fixed on a stereotaxic apparatus (Narishige). A 4-mm-length stainless-steel guide cannula (Eicom, San Diego, CA) was implanted into the PBN (AP, −5.2 mm; LR, +1.7 mm; DV, −2.4 mm from bregma). The guide cannula was attached to the skull with dental cement, and a dummy cannula (Eicom) was inserted into the guide cannula. On the day of the experiment, the dummy cannula was replaced with injection cannula (Eicom) connected with a 10-μl Hamilton syringe mounted on a micro-infusion pump (Eicom). Muscimol (2 mM, Sigma-Aldrich) or saline was infused for 1 min at a rate of 0.25 μL/min. 2MT was presented 15 min after the infusions and cutaneous temperature was measured for 20 min.

**Statistics and reproducibility**. GraphPad Prism8 was used for statistical analysis. Values are expressed as mean ± standard error of mean. Methods used for statistical analysis are indicated in Supplementary Data 1 file and in the figure legends.

**Reporting summary**. Further information on research design is available in the Nature Research Reporting Summary linked to this article.

## Data availability

All data used in the analysis are available in the main text, in the Supplementary Information, and in the Supplementary Data 1 file. All relevant data are available from K. K. (kobayakk@hirakata.kmu.ac.jp) upon request.

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

## Acknowledgements

We are grateful to Dr. Kiichi Hirota for providing suggestions regarding the analysis of anti-hypoxia effects. We are grateful to Drs. Shigetada Nakanishi and Tatsuo Kinashi for critical comments on the manuscript. We thank Dai Kanagawa and Aiko Yasuda for providing technical assistance. This work was supported by the following foundations: JSPS KAKENHI (16K07445 to T.M.; 16H06142 to T.I.; 20H04849, 18H02546, 17H05586, and 16K14558 to R.K.; 20K20578, 18K19350, and 18H04806, 16H02591 to K.K.); the Japan Science and Technology Agency, A-STEP grant (to R.K.); the Takeda Science Foundation (to T.M., T.I., R.K. and K.K.); the Canon Foundation (to K.K.); the Dai-ichi Sankyo Foundation (to R.K. and K.K.); the Naito Foundation (to T.I. and K.K.); the Sumitomo Foundation (to K.K.); the Uehara Foundation (to K.K.); the Asahi Glass Foundation (to K.K.); the Terumo Foundation (to K.K.); Mishima Kaiun Memorial Foundation (to T.M.); and AMED-CREST (JP18gm010003 (to T.S.).

## Author contributions

K.K. designed the study and experiments; K.K. wrote the manuscript with R.K.; T.I. performed the experiments described in Figs. 1 and 2 with L.T. and T.M.; T.M. performed the experiments described in Fig. 3 with L.T. and T.S.; T.M. performed the experiments described in Figs. 4 and 6 with L.T., R.K., and K.K.; R.K. performed the experiment described in Fig. 5 with K.K. and T. I.

## Competing interests

The authors declare no competing interests.
