## [Peer Review File · Communications Biology]

Reviewers' comments:

Reviewer #1 (Remarks to the Author):

The authors in this manuscript describe a hibernation-like state in mice, characterized by reduced body temperature and metabolism, induced by a synthetic odor (2MT). 2MT is more powerful than natural odors in causing fear/freezing. The authors show that this state mitigates the consequences of hypoxia and ischemia on the brain and immune system. Then the authors map the circuit and chemogenetically elicit a similar condition by activating the nucleus of the solitary tract (NST). Overall, the work is intriguing and comprehensive and provides a circuit level understanding of the hibernation-like state. However, it is not clear if the odor induced state is novel or related to those previously reported. For example, two recent mechanistic studies on this topic (torpor and hibernation-like state) have to be discussed, especially because they implicate other brain regions in hibernation state.

S. Hrvatin et al., "Neurons that regulate mouse torpor," *Nature*, doi:10.1038/s41586-020-2387-5, 2020.

T.M. Takahashi et al., "A discrete neuronal circuit induces a hibernation-like state in rodents," *Nature*, doi:10.1038/s41586-020-2163-6, 2020.

The uniqueness of odor-induced hypothermia in this report vs. hypothermia induced by other chemicals or by the manipulation of ambient temperature and feeding in published reports is not clear either. Again, integration of the current findings into existing knowledge in this area would be useful. The sheer amount of data is a little overwhelming, prioritization of data to focus on the main message would make the paper easier to comprehend. The authors initially emphasize the relevance of the behavioral state to fear, then move on to promote its significance in mitigating hypoxia-induced changes in the brain, then they turn to effects in the immune system, then switch to metabolism, and finally focus on the circuit. Although all of these data are connected to the phenomenon, a more linear story would increase the impact of the work. Less developed/relevant portions of the paper (for example those with immune related effects) could be deleted or moved to the supplement.

Specific points that could improve the manuscript are listed below.

1. The abstract is rambling, it does not outline the question and problem and the main direction of the research well. The authors intention and the main direction of the studies become clear only by reading the entire introduction. A major rewriting of the abstract would be useful.
2. The idea of fear-induced hypothermia seems to be contradictory to the general understanding of fear/anxiety, as fear typically increases heart rate and body temperature. Is the freezing component of the 2MT effect independent of its hypothermia effect? How does the fear circuit interact with the hypothermia circuit? The authors try to explain this issue by distinguishing between innate and conditioned fear in terms of temperature regulation (referring to their 2015 work), but a more detailed explanation in the discussion section would be needed. For example, why would hypothermia and reduced energy expenditure in innate fear but hyperthermia and increased energy expenditure in conditioned fear be advantageous for the individual?
3. In the result section the authors show an impressive reduction of core temperature by 2MT that can reach 10oC for hours by the continuous presence of the chemical. It would be useful to show activity in this torpor like condition. Locomotor activity was measured only after the removal of the chemical.
4. Then the authors show that the 2MT-induced reduction in core temperature is likely due to a reduction in oxygen consumption and metabolism in general, rather than reduced brown fat activity or increased heat dissipation. These experiments are well done and use a wide range of techniques including genetically modified mice. This line of work was expanded to metabolism, measuring metabolic flux with ¹³C glucose. This experiment showed increased glycolysis and reduced TCA cycle activity in the brain. Increased glycolysis is counterintuitive given reduced oxygen consumption, that needs explanation in the discussion. Specifically, how to explain increased fructose 6P but reduced fructose 1,6 biphosphate levels and similarly, increased lactate but reduced levels of its metabolite

pyruvate?

5. Although the immunological consequences of 2MT is interesting and could be important in neuroprotection, it distracts from the brain and brain ischemia focus of the paper. Indeed, cytokine responses are not directly linked to the brain. How are these peripheral responses relevant to increased hypoxia tolerance in the model. Mechanistically, it is also unclear how 2MT elicits these peripheral effects.

6. The last section of the results attempts to identify the 2MT-induced neuronal pathway that elicits and coordinates the brain protection mechanism. Although it is reasonable to focus on the brainstem, and to show chemical induced fos expression in these regions, whole brain expression of fos would be needed to judge whether these locations are indeed prominent relative to other regions. Active vs inactive chemical-related neuronal activation would be particularly useful.

7. Finally, the authors show that NTS activation by DREADD increases freezing and reduces body temp and oxygen consumption. It would be nice to show that inhibitory DREADD blocks 2MT effects. This would show the specificity of this pathway in brain protection. Also, it is not clear why CNO increases freezing even in control mCherry animals.

Minor

1. Why are multiple pictures of PBN and SC while singular representations of other nuclei shown in Fig.

6. It seems that GFP is from the Allen atlas while fos from the authors experiments.

2. In addition to the Abstract, the Introduction could be improved by eliminating repetitive sentences.

Dramatic statements, such as "Response determines life and death" should be tuned down.

Anthropomorphizing should also be avoided: "fear of death because of lack of access to food and water". It is also unusual to use wording such as "innate cold and learned warm fear".

3. References are listed in the text by names but sometimes by number.

Reviewer #2 (Remarks to the Author):

Thiazoline-related fear odors (tFOs) -TRPA1 agonists induce hibernation-like hypothermia/hypometabolism

In general, the paper describes results from a comprehensive study describing unique effects of novel tFOs through a proposed TRPA1 mechanism. The authors liken the effects to hibernation, but provide insufficient evidence to connect these phenomena. Clearly, 2MT suppresses oxygen consumption and the parallel decrease in core and cutaneous temperature is consistent with metabolic suppression. The effect, however, is accompanied by a physiologically significant decrease in O₂ saturation. The phenomenon and neural pathways involved describe a unique and exaggerated fear-like response. As such the phenomenon is of interest without over-stating its relationship to hibernation.

The idea that fear perception produces life-protective metabolism is a novel concept, however, distinguishing the concept from current understanding of the adaptive significance of fear responses could be developed in more detail. Is the innate fear observed, an exaggeration of fear-induced freezing?

Abstract:

The abstract describes the author's take home interpretation of findings without any reference to a gap in knowledge, hypothesis, approach or results.

Introduction:

There is no discussion of what is known, what is not known, the gap in knowledge and how the study was designed to fill this gap. Hence, it is not possible to assess the rigor of the prior science that

frames the need for more study.

The authors do provide a description of their own prior work and suggest that while characterizing effects of thiazoline-related fear odors developed in their own earlier work, they observed enhanced innate fear behaviors. Here they describe those observations.

Results

Fig. 1: interesting decrease in core Tb that follows decrease in cutaneous Temp.

Fig. 2: O₂ saturation data suggests inadequate perfusion is at the heart of metabolic suppression and decrease in Tb. Metabolic suppression as seen in hibernation or torpor maintains O₂ saturation despite bradycardia and decreased perfusion pressure.

Nonetheless, there is no evidence of a hypoxic ventilatory response at onset of odor that would be expected if there was an initial period of hypoxia. Breath rate decreases, but so does O₂ saturation. Why is sO₂ not decreased in the no odor group during 4% hypoxia? Why is there not a classic HVR in the no odor group when exposed to 4% O₂?

Figure 3:

Overall, interpretation of incorporation of ¹³C glucose does not convey an informed understanding of brain metabolism. Oxygen is not utilized in the TCA cycle. Oxygen is used during oxidative phosphorylation. ¹³C in blood will be present in brain.

Figure 4: which fold changes are significantly different? If only changes that were sig dif are shown, this needs to be specified in the figure caption.

Fig. 5: Is it novel to show that decreasing Tb protects the brain from I/R injury?

Fig 6. Define abbreviations for brain regions used in the figure. Areas of c-fos expression induced by 4E2MT are not consistent with areas active in hibernation (PMID: 17912746)

Discussion

The discussion relies entirely on a mis-informed and over- interpreted connection with hibernation. 2MT increases glucose incorporation and glycolysis and TCA cycle is downregulated. These effects are inconsistent with hibernation.

No text is devoted to relating the findings to what is currently known about metabolic responses to fear or to caveats to the approach or interpretation.

Reviewer #3 (Remarks to the Author):

In this study, the authors' team investigates the effects of 2MT on the "central crisis pathway" which helps to survive extreme threats. The authors demonstrate very robust effects of 2MT exposure on a variety of physiological, cellular & molecular measures which all helps to survive life-threatening situations such as hypoxia, ischemia, sepsis, etc.

I congratulate the authors' team for this impressive study. I enjoyed reading this manuscript a lot. Actually, I don't have any major comments to the manuscript but only a few minor comments (see below). In general, I found the result and discussion section very clear and convincing, whereas my feeling is that the summary and the introduction could benefit from a bit editing. This refers mainly to the 'flow' and order of arguments which are, in my opinion, a bit different in these two manuscript parts which was confusing for me.

Further minor comments:

(1) The "Wang et al., 2018b" citation on p.6 should be changed into a superscript number.

(2) Page 7: I don't understand the term "tFOs technology". Is this simply exposure to tFOs?

(3) Page 8: The first sentence of the last paragraph is irritating. LiCl is the US. The CR is later induced by CS, not by the US. I would rather argue that LiCl as an US induces an UR, i.e. an innate fear/stress response. In this context, it might be good to consider the use of "fear/stress" instead of only "fear".

(4) Figure 1: Consider adding the duration of restraint in the caption.

(5) Page 27: I don't understand why 2MT was administered i.p. in this experiment. Why no odor exposure? Especially since the authors later discuss that the sensory activation of fFO activates the central crisis pathway.

(6) Do the authors know whether ischemia/reperfusion lesions are decreased if 2Mt exposure is after inducing ischemia?

Reviewers' comments:

Reviewer #1 (Remarks to the Author):

The authors in this manuscript describe a hibernation-like state in mice, characterized by reduced body temperature and metabolism, induced by a synthetic odor (2MT). 2MT is more powerful than natural odors in causing fear/freezing. The authors show that this state mitigates the consequences of hypoxia and ischemia on the brain and immune system. Then the authors map the circuit and chemogenetically elicit a similar condition by activating the nucleus of the solitary tract (NST). Overall, the work is intriguing and comprehensive and provides a circuit level understanding of the hibernation-like state. However, it is not clear if the odor induced state is novel or related to those previously reported. For example, two recent mechanistic studies on this topic (torpor and hibernation-like state) have to be discussed, especially because they implicate other brain regions in hibernation state. S. Hrvatin et al., "Neurons that regulate mouse torpor," *Nature*, doi:10.1038/s41586-020-2387-5, 2020.

T.M. Takahashi et al., "A discrete neuronal circuit induces a hibernation-like state in rodents," *Nature*, doi:10.1038/s41586-020-2163-6, 2020.

The uniqueness of odor-induced hypothermia in this report vs. hypothermia induced by other chemicals or by the manipulation of ambient temperature and feeding in published reports is not clear either. Again, integration of the current findings into existing knowledge in this area would be useful. The sheer amount of data is a little overwhelming, prioritization of data to focus on the main message would make the paper easier to comprehend. The authors initially emphasize the relevance of the behavioral state to fear, then move on to promote its significance in mitigating hypoxia-induced changes in the brain, then they turn to effects in the immune system, then switch to metabolism, and finally focus on the circuit. Although all of these data are connected to the phenomenon, a more linear story would increase the impact of the work. Less developed/relevant portions of the paper (for example those with immune related effects) could be deleted or moved to the supplement.

Specific points that could improve the manuscript are listed below.

1. The abstract is rambling, it does not outline the question and problem and the main direction of the research well. The authors intention and the main direction of the studies become clear only by reading the entire introduction. A major rewriting of the abstract would be useful.
2. The idea of fear-induced hypothermia seems to be contradictory to the general

understanding of fear/anxiety, as fear typically increases heart rate and body temperature. Is the freezing component of the 2MT effect independent of its hypothermia effect? How does the fear circuit interact with the hypothermia circuit? The authors try to explain this issue by distinguishing between innate and conditioned fear in terms of temperature regulation (referring to their 2015 work), but a more detailed explanation in the discussion section would be needed. For example, why would hypothermia and reduced energy expenditure in innate fear but hyperthermia and increased energy expenditure in conditioned fear be advantageous for the individual?

3. In the result section the authors show an impressive reduction of core temperature by 2MT that can reach 10°C for hours by the continuous presence of the chemical. It would be useful to show activity in this torpor like condition. Locomotor activity was measured only after the removal of the chemical.

4. Then the authors show that the 2MT-induced reduction in core temperature is likely due to a reduction in oxygen consumption and metabolism in general, rather than reduced brown fat activity or increased heat dissipation. These experiments are well done and use a wide range of techniques including genetically modified mice. This line of work was expanded to metabolism, measuring metabolic flux with ¹³C glucose. This experiment showed increased glycolysis and reduced TCA cycle activity in the brain. Increased glycolysis is counterintuitive given reduced oxygen consumption, that needs explanation in the discussion. Specifically, how to explain increased fructose 6P but reduced fructose 1,6 biphosphate levels and similarly, increased lactate but reduced levels of its metabolite pyruvate?

5. Although the immunological consequences of 2MT is interesting and could be important in neuroprotection, it distracts from the brain and brain ischemia focus of the paper. Indeed, cytokine responses are not directly linked to the brain. How are these peripheral responses relevant to increased hypoxia tolerance in the model. Mechanistically, it is also unclear how 2MT elicits these peripheral effects.

6. The last section of the results attempts to identify the 2MT-induced neuronal pathway that elicits and coordinates the brain protection mechanism. Although it is reasonable to focus on the brainstem, and to show chemical induced fos expression in these regions, whole brain expression of fos would be needed to judge whether these locations are indeed prominent relative to other regions. Active vs inactive chemical-related neuronal activation would be particularly useful.

7. Finally, the authors show that NTS activation by DREADD increases freezing and reduces body temp and oxygen consumption. It would be nice to show that inhibitory DREADD blocks 2MT effects. This would show the specificity of this pathway in brain protection. Also, it is not

clear why CNO increases freezing even in control mCherry animals.

Minor

1. Why are multiple pictures of PBN and SC while singular representations of other nuclei shown in Fig. 6. It seems that GFP is from the Allen atlas while fos from the authors experiments.
2. In addition to the Abstract, the Introduction could be improved by eliminating repetitive sentences. Dramatic statements, such as "Response determines life and death" should be tuned down. Anthropomorphizing should also be avoided: "fear of death because of lack of access to food and water". It is also unusual to use wording such as "innate cold and learned warm fear".
3. References are listed in the text by names but sometimes by number.

Reviewer #2 (Remarks to the Author):

Thiazoline-related fear odors (tFOs) -TRPA1 agonists induce hibernation-like hypothermia/hypometabolism

In general, the paper describes results from a comprehensive study describing unique effects of novel tFOs through a proposed TRPA1 mechanism. The authors liken the effects to hibernation, but provide insufficient evidence to connect these phenomena. Clearly, 2MT suppresses oxygen consumption and the parallel decrease in core and cutaneous temperature is consistent with metabolic suppression. The effect, however, is accompanied by a physiologically significant decrease in O₂ saturation. The phenomenon and neural pathways involved describe a unique and exaggerated fear-like response. As such the phenomenon is of interest without over-stating its relationship to hibernation. The idea that fear perception produces life-protective metabolism is a novel concept, however, distinguishing the concept from current understanding of the adaptive significance of fear responses could be developed in more detail. Is the innate fear observed, an exaggeration of fear-induced freezing?

Abstract:

The abstract describes the author's take home interpretation of findings without any reference to a gap in knowledge, hypothesis, approach or results.

Introduction:

There is no discussion of what is known, what is not known, the gap in knowledge and how the study was designed to fill this gap. Hence, it is not possible to assess the rigor of the prior science that frames the need for more study.

The authors do provide a description of their own prior work and suggest that while characterizing effects of thiazoline-related fear odors developed in their own earlier work, they observed enhanced innate fear behaviors. Here they describe those observations.

Results

Fig. 1: interesting decrease in core Tb that follows decrease in cutaneous Temp. Fig. 2: O₂ saturation data suggests inadequate perfusion is at the heart of metabolic suppression and decrease in Tb. Metabolic suppression as seen in hibernation or torpor maintains O₂ saturation despite bradycardia and decreased perfusion pressure. Nonetheless, there is no evidence of a hypoxic ventilatory response at onset of odor that would be expected if there was an initial period of hypoxia. Breath rate decreases, but so does O₂ saturation.

Why is sO₂ not decreased in the no odor group during 4% hypoxia? Why is there not a classic HVR in the no odor group when exposed to 4% O₂?

Figure 3:

Overall, interpretation of incorporation of ¹³C glucose does not convey an informed understanding of brain metabolism. Oxygen is not utilized in the TCA cycle. Oxygen is used during oxidative phosphorylation. ¹³C in blood will be present in brain.

Figure 4: which fold changes are significantly different? If only changes that were sig dif are shown, this needs to be specified in the figure caption.

Fig. 5: Is it novel to show that decreasing Tb protects the brain from I/R injury?

Fig 6. Define abbreviations for brain regions used in the figure. Areas of c-fos expression induced by 4E2MT are not consistent with areas active in hibernation (PMID: 17912746)

Discussion

The discussion relies entirely on a mis-informed and over- interpreted connection with hibernation.

2MT increases glucose incorporation and glycolysis and TCA cycle is downregulated. These effects are inconsistent with hibernation.

No text is devoted to relating the findings to what is currently known about metabolic responses to fear or to caveats to the approach or interpretation.

Reviewer #3 (Remarks to the Author):

In this study, the authors' team investigates the effects of 2MT on the "central crisis pathway" which helps to survive extreme threats. The authors demonstrate very robust effects of 2MT exposure on a variety of physiological, cellular & molecular measures which all helps to survive life-threatening situations such as hypoxia, ischemia, sepsis, etc.

I congratulate the authors' team for this impressive study. I enjoyed reading this manuscript a lot. Actually, I don't have any major comments to the manuscript but only a few minor comments (see below). In general, I found the result and discussion section very clear and convincing, whereas my feeling is that the summary and the introduction could benefit from a bit editing. This refers mainly to the 'flow' and order of arguments which are, in my opinion, a bit different in these two manuscript parts which was confusing for me.

Further minor comments:

- (1) The "Wang et al., 2018b" citation on p.6 should be changed into a superscript number.
- (2) Page 7: I don't understand the term "tFOs technology". Is this simply exposure to tFOs?
- (3) Page 8: The first sentence of the last paragraph is irritating. LiCl is the US. The CR is later induced by CS, not by the US. I would rather argue that LiCl as an US induces an UR, i.e. an innate fear/stress response. In this context, it might be good to consider the use of "fear/stress" instead of only "fear".
- (4) Figure 1: Consider adding the duration of restraint in the caption.
- (5) Page 27: I don't understand why 2MT was administered i.p. in this experiment. Why no odor exposure? Especially since the authors later discuss that the sensory activation of fFO activates the central crisis pathway.
- (6) Do the authors know whether ischemia/reperfusion lesions are decreased if 2Mt exposure is after inducing ischemia?

Responses to reviewers:

Reviewer #1 (Remarks to the Author):

The authors in this manuscript describe a hibernation-like state in mice, characterized by reduced body temperature and metabolism, induced by a synthetic odor (2MT). 2MT is more powerful than natural odors in causing fear/freezing. The authors show that this state mitigates the consequences of hypoxia and ischemia on the brain and immune system. Then the authors map the circuit and chemogenetically elicit a similar condition by activating the nucleus of the solitary tract (NST). Overall, the work is intriguing and comprehensive and provides a circuit level understanding of the hibernation-like state. However, it is not clear if the odor induced state is novel or related to those previously reported. For example, two recent mechanistic studies on this topic (torpor and hibernation-like state) have to be discussed, especially because they implicate other brain regions in hibernation state.

S. Hrvatin et al., "Neurons that regulate mouse torpor," *Nature*, doi:10.1038/s41586-020-2387-5, 2020.

T.M. Takahashi et al., "A discrete neuronal circuit induces a hibernation-like state in rodents," *Nature*, doi:10.1038/s41586-020-2163-6, 2020.

Response: Thank you for your numerous crucial and informative comments. We believe that complying with the suggestions has helped us to present the manuscript's main argument more clearly. As reviewer 1 has suggested, it is important to clearly state the relationship between the physiological states induced by 2MT and hibernation/torpor. As reviewer 2 has pointed out, our results indicate that hibernation/torpor and 2MT-induced states have differences in brain metabolism, oxygen saturation, and brain activity. Further, we recently performed brain-mapping experiments, revealing the differences in brain activity between hibernation/torpor state and 2MT-induced survival fate. The results are shown in the newly added Figure 5. These analyses have allowed us to identify similarities and differences between torpor/hibernation and the physiological states induced by 2MT. We have also cited recently published studies suggested by Reviewer 1, as well a study published in *Cell* in 2016 that reported similar experimental results (lines 40-46).

The uniqueness of odor-induced hypothermia in this report vs. hypothermia induced by other chemicals or by the manipulation of ambient temperature and feeding in published

reports is not clear either. Again, integration of the current findings into existing knowledge in this area would be useful.

Response: The similarities and differences between 2MT-induced hypothermia and other known hypothermia-inducing phenomena have been clearly explained in the revised manuscript.

The phenomenon of inhalation of low concentrations of hydrogen sulfide or administration of 2-deoxy-glucose is known to induce hypothermia. The administration of these compounds and 2MT stimulation induce hypothermia presumably by different mechanisms. This has been described in lines 179-194 and lines 309-319 in the revised manuscript. Further, to clarify the difference between hydrogen sulfide-induced hypothermia and 2MT-induced hypothermia, a description of the difference in oxygen saturation has been added in lines 315-316.

To clearly characterize 2MT-induced hypothermia compared to known hypothermia-induced phenomena, we have added a new paragraph, "Unique features of 2MT-induced hypothermia," to the results section (lines 308-363), which includes the results of c-fos mapping analysis for seven brain regions involved in the regulation of torpor/hibernation.

The sheer amount of data is a little overwhelming, prioritization of data to focus on the main message would make the paper easier to comprehend. The authors initially emphasize the relevance of the behavioral state to fear, then move on to promote its significance in mitigating hypoxia-induced changes in the brain, then they turn to effects in the immune system, then switch to metabolism, and finally focus on the circuit. Although all of these data are connected to the phenomenon, a more linear story would increase the impact of the work. Less developed/relevant portions of the paper (for example those with immune related effects) could be deleted or moved to the supplement.

Response: According to the suggestion of Reviewer 1, we have deleted the experimental data of the immune-related effects to focus on the central message of this study. Further, we performed a new set of experiments to analyze 2MT-induced *c-fos* expression in various brain regions, which are activated in hibernating animals. The results clearly show the differences between 2MT-induced hypothermia and hibernation/torpor.

Specific points that could improve the manuscript are listed below.

1. The abstract is rambling, it does not outline the question and problem and the main direction of the research well. The authors intention and the main direction of the studies

become clear only by reading the entire Introduction. A major rewriting of the abstract would be useful.

Response: We have rewritten the abstract to clearly outline the question and the main aim of the research. Further, we have mentioned the differences between the 2MT-induced state and the hibernation/torpor state in the revised Abstract.

2. The idea of fear-induced hypothermia seems to be contradictory to the general understanding of fear/anxiety, as fear typically increases heart rate and body temperature. Is the freezing component of the 2MT effect independent of its hypothermia effect? How does the fear circuit interact with the hypothermia circuit? The authors try to explain this issue by distinguishing between innate and conditioned fear in terms of temperature regulation (referring to their 2015 work), but a more detailed explanation in the discussion section would be needed. For example, why would hypothermia and reduced energy expenditure in innate fear but hyperthermia and increased energy expenditure in conditioned fear be advantageous for the individual?

Response: A new paragraph has been added in the Discussion section to address this point (lines 437-454). As the Reviewer indicated, it is generally accepted that fear stimuli increase body temperature and heart rate. On the other hand, this study demonstrates that innate fear elicited 2MT induced hypothermia and bradycardia. Thus, the addition of the paragraph will improve the reader's understanding of the implications of different physiological responses induced by different types of fear stimuli.

3. In the result section the authors show an impressive reduction of core temperature by 2MT that can reach 10oC for hours by the continuous presence of the chemical. It would be useful to show activity in this torpor like condition. Locomotor activity was measured only after the removal of the chemical.

Response: We recently analyzed locomotor activity during long exposure to 2MT odorant. The results of this experiment are shown in Figure S1 and have been described in the revised manuscript (lines 106-108).

4. Then the authors show that the 2MT-induced reduction in core temperature is likely due to a reduction in oxygen consumption and metabolism in general, rather than reduced brown fat activity or increased heat dissipation. These experiments are well done and use a wide range of techniques including genetically modified mice. This line of work was expanded to metabolism, measuring metabolic flux with ¹³C glucose. This experiment showed increased glycolysis and reduced TCA cycle activity in the brain. Increased glycolysis is counterintuitive given reduced oxygen consumption, that needs explanation in the discussion. Specifically, how to explain increased fructose 6P but reduced fructose 1,6 biphosphate levels and similarly, increased lactate but reduced levels of its metabolite pyruvate?

Response: As Reviewer 1 has pointed out, upregulated glycolysis in the brain during reduced oxygen consumption is important. We have included this point in the Discussion section of the revised manuscript (lines 476-491). The metabolomic analysis shows a decrease in the amount of fructose 1,6 bisphosphate. This result can be interpreted in two contradictory ways—the decrease in the amount of fructose 1,6 bisphosphate could be due to either decelerated or accelerated glycolysis. Thus, metabolome analysis alone cannot determine whether glycolysis was accelerated or decelerated. In order to distinguish between these two possibilities, a metabolome flux analysis is necessary. In our metabolome flux analysis, only six metabolites involved in glycolysis and the TCA cycle were detected. It has been pointed out that pyruvate decreases. However, pyruvate was not detected during either metabolome analysis or metabolic flux analysis. Therefore, it is not clear whether pyruvate is reduced during 2MT stimulation. We have added a sentence to clarify which metabolites were detectable and which were not in the revised Results section (lines 215-217).

5. Although the immunological consequences of 2MT is interesting and could be important in neuroprotection, it distracts from the brain and brain ischemia focus of the paper. Indeed, cytokine responses are not directly linked to the brain. How are these peripheral responses relevant to increased hypoxia tolerance in the model. Mechanistically, it is also unclear how 2MT elicits these peripheral effects.

Response: As mentioned above, according to the advice of Reviewer 1, we have deleted the data for immune-related effects induced by 2MT in the revised manuscript. Instead, in order to focus on the mechanisms of hypothermia/hypometabolism induced by 2MT, we have analyzed *c-fos* mRNA

expression in the brain regions involved in hibernation/torpor as shown in new Figure 5. A new subsections describing the differences between 2MT-induced hypothermia and hibernation/torpor in terms of brain activities has also been added. (lines 308-363).

6. The last section of the results attempts to identify the 2MT-induced neuronal pathway that elicits and coordinates the brain protection mechanism. Although it is reasonable to focus on the brainstem, and to show chemical induced fos expression in these regions, whole brain expression of fos would be needed to judge whether these locations are indeed prominent relative to other regions. Active vs inactive chemical-related neuronal activation would be particularly useful.

Response: It is crucial to justify the reason for focusing on the brainstem area in this study. According to the comment of Reviewer 1, we performed a *c-fos* brain mapping using different types of odorants. Three types of odor stimulation conditions were used: no odor condition, learned fear odor stimulation, and innate fear odor stimulation (2MT). Six brain regions known to be involved in hibernation/torpor were included in the analysis. Specifically, medial preoptic area (MPA), paraventricular nucleus of hypothalamus (PVN), suprachiasmatic nucleus (SCN), and reticular thalamic nucleus (Rt), choroid plexus (Chp), and tanycytes (Ta) were selected. The results are shown in the new Figure 5. From this analysis, we could not detect brain regions that were activated only by 2MT stimulation that induced hypothermia.

We previously showed that 2MT induces fear-related behaviors by activating TRPA1 in the trigeminal nerve, which transmits information to the Sp5 in the brainstem area. Therefore, it is possible that the information transmitted to the brainstem may play an important role with respect to 2MT-induced physiological responses. *Trpa1* is expressed not only in the trigeminal nerve but also in the vagus nerve. Correspondingly, 2MT stimulation induced the expression of *c-fos* mRNA in the Sp5, which receives axonal projection by the trigeminal nerve, as well as in the NST, which receives axonal projection from the vagus nerve (Figure S5). Therefore, 2MT stimulation may induce hypothermia and anti-hypoxia by activating neural pathways that originate from these brainstem areas. Therefore, in the present study, we aimed to elucidate the neural pathway originating from these brainstem areas.

In the revised manuscript, we have added the results of these experiments and the explanation of why we focused on these brainstem areas (lines 320-363).

7. Finally, the authors show that NTS activation by DREADD increases freezing and reduces

body temp and oxygen consumption. It would be nice to show that inhibitory DREADD blocks 2MT effects. This would show the specificity of this pathway in brain protection. Also, it is not clear why CNO increases freezing even in control mCherry animals.

Response: Basically, it is desirable to present the Gi-DREADD in addition to the Gq-DREADD results. Thus, we performed experiments to inactivate the NST-PBN pathway using Gi-DREADD. The results of the experiments are shown in new Figure S8A-B. It was anticipated that Gi-DREADD might inhibit the induction of hypothermia caused by 2MT, but in fact, no significant inhibitory effect was observed. 2MT activates at least the Sp5 and NST in the brainstem area. As shown in Figure 6A and B, the PBN receives axonal projections from the Sp5 in addition to the NST. Therefore, it is presumed that the inhibition of the NST-PBN pathway alone could not significantly inhibit the effects induced by 2MT.

Since both the Sp5 and NST neurons project to the PBN, we hypothesized that inhibition of the neural activity in the PBN might inhibit 2MT-induced hypothermia. Indeed, inhibition of neural activity in the PBN by stereotaxic injection of muscimol, a GABA-A agonist, significantly inhibited 2MT-induced hypothermia (new Figure S8C-G). Our experiments collectively show that 2MT stimulation activates the neural pathway from the brainstem to the PBN, that activation of this pathway with Gq-DREADD induced hypothermia, and that inhibition of the neural activity of the PBN inhibited the 2MT-induced hypothermia. Text regarding the result, the limitations, and the interpretation of these experiments have been added in the manuscript (lines 406-412).

As Reviewer 1 indicated, indeed, injections of CNO induced weak Freezing behavior in control mice as well as in mice injected with Gq-DREADD. We hypothesize that this is a consequence of the mice being stressed by the intraperitoneal injection of the CNO solution after being captured by the experimenter. The fact that the mice were dehaired to measure body surface temperature in this experiment may have contributed to the elevated stress response. An important point is that CNO injection induced a more robust freezing behavior in Gq-DREADD mice than in control mice.

Minor

1. Why are multiple pictures of PBN and SC while singular representations of other nuclei shown in Fig. 6. It seems that GFP is from the Allen atlas while fos from the authors experiments.

Response: PBNs and SC differ from the other regions in that they receive axonal projections from both Sp5 and NST. Therefore, in Figure 6A and B, we show two photographs only for PBNs and SC.

The photographs showing *c-fos* expression in PBNs and SC are derived from two different individuals.

2. In addition to the Abstract, the Introduction could be improved by eliminating repetitive sentences. Dramatic statements, such as "Response determines life and death" should be tuned down. Anthropomorphizing should also be avoided: "fear of death because of lack of access to food and water".

It is also unusual to use wording such as "innate cold and learned warm fear".

Response: According to the advice of Reviewer 1, the following changes were made in the manuscript.

- 1) We have deleted repetitive sentences (five sentences) in the Introduction.
- 2) The expression "life or death" has been changed to "survivability" (p6) or "life protection" (p25).
- 3) "Restraint in a tight space is considered to induce innate fear because it will result in death due to lack of access to food and water" has been changed to "Restraint in a tight space is considered to induce innate fear/stress because it prevents access to food and water and may lead to death" (lines 120-121).
- 4) The subheading "Innate cold versus learned warm fear" has been replaced with "Antagonistic physiological responses between innate and learned fear" (line 92), and "innate cold fear and learned warm fear" was changed to "innate hypothermic fear state and learned hyperthermic state" (lines 127).

3. References are listed in the text by names but sometimes by number.

Response: References are all listed by number in the revised manuscript.

Reviewer #2 (Remarks to the Author):

Thiazoline-related fear odors (tFOs) -TRPA1 agonists induce hibernation-like hypothermia/hypometabolism

In general, the paper describes results from a comprehensive study describing unique effects of novel tFOs through a proposed TRPA1 mechanism. The authors liken the effects

to hibernation, but provide insufficient evidence to connect these phenomena. Clearly, 2MT suppresses oxygen consumption and the parallel decrease in core and cutaneous temperature is consistent with metabolic suppression. The effect, however, is accompanied by a physiologically significant decrease in O₂ saturation. The phenomenon and neural pathways involved describe a unique and exaggerated fear-like response. As such the phenomenon is of interest without over-stating its relationship to hibernation. The idea that fear perception produces life-protective metabolism is a novel concept, however, distinguishing the concept from current understanding of the adaptive significance of fear responses could be developed in more detail. Is the innate fear observed, an exaggeration of fear-induced freezing?

Response: Thank you for your important and thought-provoking comments. We believe that responding to the comments has allowed us to demonstrate the characteristics of the phenomenon induced by the 2MT stimulus. As Reviewer 2 has pointed out, although there are similarities between the 2MT-induced fear state and hibernation, there are clear differences between them. Thus, it is important to be careful to not overstate the relationship between 2MT-induced innate fear state and hibernation. As Reviewer 1 pointed out, it is also important to describe in more depth the relationship between the 2MT-induced innate fear state and known fear responses or hypothermia-induced phenomena. Throughout the manuscript, we have rewritten the text that was over-stating the relationship between 2MT-induced fear state and hibernation. We also conducted additional experiments to clearly show the similarities and differences between the 2MT-induced hypothermic state and the hibernation/torpor state (Figure 5). A new subsection, "Unique features of 2MT-induced hypothermia," has been added to summarize these results (lines 308-363). In addition, a new paragraph discussing the relationship between the 2MT-induced innate fear state and known fear responses has been added in the Discussion section (lines 437-454). We hope that these improvements answer many of the concerns raised by Reviewer 2.

Abstract:

The abstract describes the author's take home interpretation of findings without any reference to a gap in knowledge, hypothesis, approach or results.

Introduction:

There is no discussion of what is known, what is not known, the gap in knowledge and how the study was designed to fill this gap. Hence, it is not possible to assess the rigor of the

prior science that frames the need for more study.

The authors do provide a description of their own prior work and suggest that while characterizing effects of thiazoline-related fear odors developed in their own earlier work, they observed enhanced innate fear behaviors. Here they describe those observations.

Response: A new paragraph has been added to the Introduction (lines 40-46) to describe the previously known knowledge in more details. In addition, statements that were considered not essential have been deleted.

Results

Fig. 1: interesting decrease in core Tb that follows decrease in cutaneous Temp. Fig. 2: O₂ saturation data suggests inadequate perfusion is at the heart of metabolic suppression and decrease in Tb. Metabolic suppression as seen in hibernation or torpor maintains O₂ saturation despite bradycardia and decreased perfusion pressure. Nonetheless, there is no evidence of a hypoxic ventilatory response at onset of odor that would be expected if there was an initial period of hypoxia. Breath rate decreases, but so does O₂ saturation.

Why is sO₂ not decreased in the no odor group during 4% hypoxia? Why is there not a classic HVR in the no odor group when exposed to 4% O₂?

Response: These points made by Reviewer 2, especially regarding the difference between known hibernation and 2MT-induced hypothermia have important implications. Hibernation/torpor did not reduce O₂ saturation, but 2MT stimulation clearly reduced O₂ saturation. Under normal conditions, the decreased O₂ saturation should cause a hypoxic ventilatory response (HVR). However, in the 2MT-induced hypothermic state, the respiratory rate did not increase in spite of the decreased O₂ saturation. This suggests that 2MT stimulation suppresses HVR. In squirrels, oxygen saturation increases during the hibernation phase; however, oxygen saturation decreases during the arousal from hibernation. Thus, although 2MT stimulation induces hypothermia, it may induce a physiological state closer to the arousal from hibernation stage than to the induction stage of hibernation.

Following Reviewer 2's suggestion, we have added sentences to the Results section to describe these important phenomena (lines 149-157).

Figure 2E shows the oxygen saturation in response to a control odor, Eugenol, which does not induce body temperature changes, under ambient air conditions and not under 4% oxygen condition, so no HVR is observed.

Figure 3:

Overall, interpretation of incorporation of ¹³C glucose does not convey an informed understanding of brain metabolism. Oxygen is not utilized in the TCA cycle. Oxygen is used during oxidative phosphorylation. ¹³C in blood will be present in brain.

Response: As Reviewer 2 has pointed out, ¹³C-glucose can be present in both the brain and the blood circulating in the brain. Therefore, there are two possibilities: ¹³C-glucose was increased in the blood or incorporation of ¹³C-glucose was increased in the brain. However, in the brain, in addition to ¹³C-glucose ¹³C-lactate was significantly increased compared to the control condition, suggesting that the incorporation of ¹³C-glucose and glycolysis is enhanced in the brain. Although there is a large amount of blood in the liver, no increase in ¹³C-glucose or ¹³C-lactate was detected (Figure S4), indicating that the enhancement in glycolysis is brain-specific. In addition, a previous study reported the similar levels of metabolites in brain extracts derived from hibernating and active meadow jumping mouse extracts as our study (Storey et al., 2011). Glycolytic metabolites such as glucose-6P and fructose-6P were markedly decreased in the hibernation phase compared to the active phase, suggesting that glycolysis is suppressed during the hibernation phase. This result is in contrast to that during 2MT-induced hypothermia; glycolysis is suppressed to save energy during hibernation, whereas during 2MT-induced crisis mode, glycolysis is enhanced probably to protect the brain from the crisis. Inhibition of glycolysis has also been reported in the brains of hibernating ground squirrels (Andrews et al., 2009). This explanation has been added in the revised manuscript (lines 215-234). As pointed out by the Reviewer, the electron transfer system requires oxygen, and the TCA circuit supplies materials to the electron transfer system and does not use oxygen. Accordingly, we have revised the manuscript (lines 200-202).

Figure 4: which fold changes are significantly different? If only changes that were sig dif are shown, this needs to be specified in the figure caption.

Response: According to the comment of Reviewer 1, we deleted the data for immune-related effects of 2MT in the revised manuscript.

Fig. 5: Is it novel to show that decreasing Tb protects the brain from I/R injury?

Response: It is well known that hypothermia has a therapeutic effect on ischemia-reperfusion injury. The present study was designed to determine whether 2MT-induced hypothermia has the same therapeutic effect. It is generally believed that increasing blood flow is effective in treating pressure ulcers, which are skin ischemia-reperfusion injury. However, in our study, 2MT stimulation induced a therapeutic effect, despite the fact that it strongly inhibited cutaneous blood flow. In addition, 2MT administration at the reperfusion stage prevented the destruction of brain tissue in the brain ischemia-reperfusion model.

Fig 6. Define abbreviations for brain regions used in the figure. Areas of *c-fos* expression induced by 4E2MT are not consistent with areas active in hibernation (PMID: 17912746)

Response: We have added abbreviations for brain regions in the legend to Figure 6 (lines 647-651). We recently performed *c-fos* mapping for brain regions active during hibernation (PMID: 17912746) and have added a subsection to describe the similarities and differences in brain activity between tFO-induced hypothermia and hibernation in the revised manuscript (lines 309-363).

Discussion

The discussion relies entirely on a mis-informed and over- interpreted connection with hibernation.

2MT increases glucose incorporation and glycolysis and TCA cycle is downregulated. These effects are inconsistent with hibernation.

No text is devoted to relating the findings to what is currently known about metabolic responses to fear or to caveats to the approach or interpretation.

Response: According to the Reviewer's comment, we have added a description of the differences between 2MT-induced hypothermia and hibernation (lines 497-499). We also discussed the relationship between 2MT-induced physiological responses and known physiological responses caused by fear in the revised manuscript (lines 437-454).

Reviewer #3 (Remarks to the Author):

In this study, the authors' team investigates the effects of 2MT on the "central crisis pathway" which helps to survive extreme threats. The authors demonstrate very robust effects of 2MT exposure on a variety of physiological, cellular & molecular measures which all helps to survive life-threatening situations such as hypoxia, ischemia, sepsis, etc.

I congratulate the authors' team for this impressive study. I enjoyed reading this manuscript a lot. Actually, I don't have any major comments to the manuscript but only a few minor comments (see below). In general, I found the result and discussion section very clear and convincing, whereas my feeling is that the summary and the Introduction could benefit from a bit editing. This refers mainly to the 'flow' and order of arguments which are, in my opinion, a bit different in these two manuscript parts which was confusing for me.

Response: Thank you for your appreciation of our manuscript. According to the advice of Reviewer 3, we have rewritten the Abstract and the Introduction to clearly outline the aim of the study, existing knowledge in the field, and the research's main direction. We hope that we were able to improve our manuscript.

Further minor comments:

(1) The "Wang et al., 2018b" citation on p.6 should be changed into a superscript number.

Response: The "Wang et al., 2018b" has been listed by number in the revised manuscript.

(2) Page 7: I don't understand the term "tFOs technology". Is this simply exposure to tFOs?

Response: According to the comment, we have changed to "tFOs" in the revised manuscript (line 87).

(3) Page 8: The first sentence of the last paragraph is irritating. LiCl is the US. The CR is later induced by CS, not by the US. I would rather argue that LiCl as an US induces an UR, i.e. an innate fear/stress response. In this context, it might be good to consider the use of "fear/stress" instead of only "fear".

Response: According to the Reviewer's advice, we have changed the sentence to " In addition to electric shock, intraperitoneal (IP) injection of lithium chloride (LiCl) can also be used as an

unconditioned stimulus²⁷. Thus, IP injection of LiCl may also elicit innate fear/stress responses.”
(line 114-116).

(4) Figure 1: Consider adding the duration of restraint in the caption.

Response: According to the Reviewer's advice, the duration of restraint has been added in the Figure Legend in the revised manuscript (lines 551-552).

(5) Page 27: I don't understand why 2MT was administered i.p. in this experiment. Why no odor exposure? Especially since the authors later discuss that the sensory activation of fFO activates the central crisis pathway.

Response: We have added our reason to use IP injection for the BCCAO experiment in lines 298-301. Both odor exposure and IP injection induced c-fos expression in the Sp5 and NST, which receives afferent inputs from the trigeminal and vagus nerves.

(6) Do the authors know whether ischemia/reperfusion lesions are decreased if 2Mt exposure is after inducing ischemia?

Response: This point made by Reviewer 3 is an important one. We showed a protective effect of 2MT when administered before ischemia in a cutaneous ischemia-reperfusion model (Figure 4A-F). We further analyzed the protective effect of 2MT in a more clinical model of the brain ischemia-reperfusion model. In this model, cerebral ischemia was induced, and reperfusion of blood flow was initiated 30 min later. Thus, we showed that 2MT administration during this reperfusion phase still has a protective effect against brain damage (Figure 4G-L).

REVIEWERS' COMMENTS:

Reviewer #1 (Remarks to the Author):

The authors were highly responsive to previous comments. They included new data, suggested by the reviewers, simplified the message and made the flow much better. The difference between hibernation and the 2MT-induced hypothermia condition is now well described. I am satisfied with the new manuscript with a few exceptions.

1. The title is still difficult to understand. What does "orchestrate hibernation and survival fate" mean? I am concerned that the reader will not be able to comprehend the topic from the title and would miss on an otherwise fascinating paper.

2. The abstract is still difficult to follow because the actual condition induced by 2MT is not well defined. I find the "survival fate" definition ungainly. My understanding is that the 2MT response is simply an evolutionary DEFENSIVE RESPONSE (death feigning), albeit exaggerated because the identified chemical elicits a supernormal response. The authors should also comment on if the system and response is conserved in human and if 2MT treatment could elicit a similar response in human. This would indicate clinical utility.

3. It is still puzzling how fructose 1,6 bisphosphate is reduced while fructose 6P and lactate levels are increase when PHD activity is diminished. There might be other interacting pathways that could be mentioned.

Reviewer #3 (Remarks to the Author):

My comments to the previous version of the manuscript are well addressed in this revised version. The manuscript clearly gained quality and can be published in its present form.

Point-by-Point Response to Reviewers' Comments

Reviewer #1 (Remarks to the Author):

The authors were highly responsive to previous comments. They included new data, suggested by the reviewers, simplified the message and made the flow much better. The difference between hibernation and the 2MT-induced hypothermia condition is now well described. I am satisfied with the new manuscript with a few exceptions.

Response: We appreciate the valuable comments made by the reviewer. As pointed out by the reviewer, the title and abstract are important for the accessibility of readers. We revised the title and abstract according to the reviewer's suggestions. We hope these changes will improve the readability and accessibility of our paper.

1. The title is still difficult to understand. What does "orchestrate hibernation and survival fate" mean? I am concerned that the reader will not be able to comprehend the topic from the title and would miss on an otherwise fascinating paper.

Response: We thank the reviewer for the comment. As pointed out by the reviewer, the title was still difficult to understand for readers. Accordingly, we changed the title to "Artificial hibernation/life-protective state induced by thiazoline-related innate fear odors." We hope this title represents the phenomena induced by innate fear odor 2MT more clearly.

2. The abstract is still difficult to follow because the actual condition induced by 2MT is not well defined. I find the "survival fate" definition ungainly. My understanding is that the 2MT response is simply an evolutionary DEFENSIVE RESPONSE (death feigning), albeit exaggerated because the identified chemical elicits a supernormal response. The authors should also comment on if the system and response is conserved in human and if 2MT treatment could elicit a similar response in human. This would indicate clinical utility.

Response: We are thankful for the valuable comment made by the reviewer. As pointed out by the reviewer, the abstract was still difficult to follow for readers. We changed "survival fate" to "physiological responses" (line 28). We agree that the evolutionary aspects of the physiological responses induced by innate fear are important. Thus, we added the following sentences to the abstract: "2MT, as a supernormal stimulus of innate fear, induced exaggerated, latent life-protective

effects in mice. If this system is preserved in humans, it may be utilized to give rise to a new field: “sensory medicine.” (lines 26-28)

3. It is still puzzling how fructose 1,6 bi phosphate is reduced while fructose 6P and lactate levels are increase when PHD activity is diminished. There might be other interacting pathways that could be mentioned.

Response: In response to the reviewer’s comment, we mentioned an alternative interpretation of our metabolomic data (lines 223-228). We also changed “glucose uptake and glycolysis” to “glucose uptake” (lines 208, 230-231, 235, 237, and 243) in the revised manuscript.

Reviewer #3 (Remarks to the Author):

My comments to the previous version of the manuscript are well adressed in this revised version. The manuscript clearly gained quality and can be published in its present form.

Response: Thank you for your positive evaluation of our manuscript. We appreciate that the comments raised by the reviewers greatly helped to greatly improve our manuscript.